# GraphEval: A Lightweight Graph-Based LLM Framework for Idea Evaluation

**Tao Feng[1]\*, Yihang Sun[2]\*, Jiaxuan You[1]**
[1]University of Illinois at Urbana-Champaign [2]Peking University
\*Equal Contribution

## Abstract

The powerful capabilities of Large Language Models (LLMs) have led to their growing use in evaluating human-generated content, particularly in evaluating research ideas within academic settings. Existing solutions primarily rely on prompt-based LLM methods or fine-tuned lightweight language models for idea evaluation. However, these methods are often unstable and struggle to comprehend the complex semantic information embedded in the ideas, impeding their ability to perform high-quality evaluations. To address the above challenges, we propose `GraphEval`, a lightweight graph-based LLM framework for idea evaluation. Our insight is that a complex idea can be broken down into comprehensible viewpoint-nodes using small prompted LLMs. These viewpoint-nodes can then be linked together through edges created from LLM-based relation extraction and/or BERT similarity scores. The created viewpoint-graph can be used to conveniently propagate scores across viewpoint-nodes to improve the robustness of the idea evaluations. In particular, we propose two lightweight graph-based methods for idea evaluation: (1) GraphEval-LP: a training-free label propagation algorithm that propagates quality labels from known viewpoint-nodes to unknown nodes; (2) GraphEval-GNN: a Graph Neural Network (GNN) that is trained to predict the quality labels given the observed graph with minimal computation resources. Moreover, to overcome LLM's limitation in objectively assessing the novelty of ideas, we further propose a novelty detection model to GraphEval-GNN to enhance its capability in judging idea novelty. Experiments on two datasets show `GraphEval` improves F1 scores by at least 14% with low computation and API costs. Additionally, `GraphEval` can effectively detect plagiarized ideas. Our codes for `GraphEval` is released at https://github.com/ulab-uiuc/GraphEval.

## 1 Introduction

With the advancement of LLMs, many tasks traditionally performed by humans, such as idea evaluations (Liang et al., 2024; Lin et al., 2023a), label annotation (Wang et al., 2024; Goel et al., 2023), or providing feedback to intelligent systems (Stamper et al., 2024; Mai et al., 2023), are now handled by LLMs. Among these applications, the use of LLMs to substitute humans in idea evaluations (Lu et al., 2024; Baek et al., 2024) carries substantial potential, where researchers can obtain much faster feedback, as well as considerable risks, where the preference and bias of LLMs could affect the development of a scientific domain. Concretely, it is well known that many reviews for paper submissions are now written with the help of LLMs, which is explicitly allowed by ICLR 2025 as well. Unfortunately, existing LLMs are often biased to be "nice and helpful" while being highly sensitive to the prompt, illustrated by Figure 1. Therefore, this paper aims to highlight a pressing research question: *how do we improve the fidelity of LLM-based idea evaluation?*

Most existing research attempts to address the problem of LLM-based idea evaluation by designing better prompt strategy (Brown, 2020; Wei et al., 2022; Wang et al., 2022; Yao et al., 2024), so that more background knowledge, feedback, or inductive bias can be incorporated to an LLM. For example, Research Agent (Baek et al., 2024) evaluates the ideas based on its five criteria added in the prompt. AI Scientist (Lu et al., 2024) introduces some prompt tricks like self-reflection (Shinn et al., 2024), providing few-shot examples (Wei et al., 2022), and response ensembling (Wang et al., 2022) to enhance the idea evaluations. However, these prompt-based evaluation methods are still

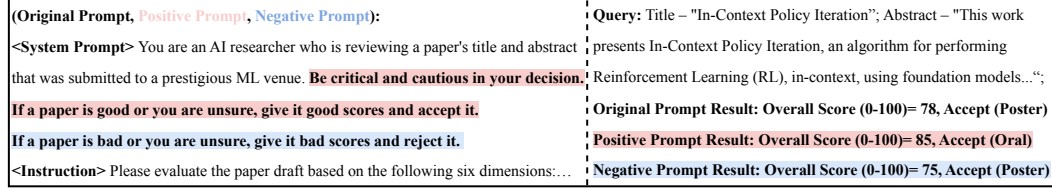

Figure 1: **Current LLMs are highly sensitive to prompts and show biases in evaluations.** This figure illustrates that even minor variations in the LLM's prompts (Original Prompt, Positive Prompt, Negative Prompt) for the same idea can lead to drastic changes in the final LLM evaluation results. Moreover, the LLM tends to always give friendly evaluations like 'Accept' and rarely gives negative evaluations such as 'Reject'. This observation demonstrates that the LLM evaluation is biased.

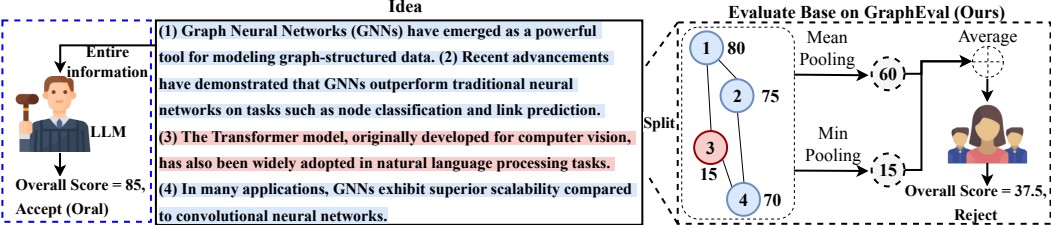

Figure 2: `GraphEval` performs a better idea evaluation than the existing LLM-based method by focusing on both the global and local information of the idea. In this figure, the part highlighted in red in the idea contain factual errors. The existing LLM-based method shown on the far left focuses solely on the global information of the idea, which often leads to overlooking factual errors interspersed within the idea. In contrast, `GraphEval` decomposes the idea into viewpoints to obtain scores for each viewpoint, then employs Mean Pooling and Min Pooling to extract global and local information of the idea, respectively. Finally, `GraphEval` derives a fair and unbiased evaluation based on these two aspects of information.

limited because: 1) they are highly sensitive to different prompts (Errica et al., 2024; Zhang et al., 2024a) and are prone to hallucinations (Sansford et al., 2024; Yao et al., 2023a); 2) they also require LLMs to possess advanced capabilities (Santu et al., 2024; Liang et al., 2024) to fully understand and judge a complex research idea, which often requires PhD-level humans in the real-world; 3) they could overlook factual inaccuracies interspersed among the ideas. As illustrated in Figure 2, existing LLM-based methods directly analyze the entire set of information, therefore easily missing the factual errors within the idea, leading to a biased evaluation.

Many studies in human psychology (Knauff & Wolf, 2010; Dijkstra et al., 2014) indicate that people often find it difficult to understand abstract ideas, which can lead to random cognition and decision-making. However, two methods can significantly enhance human understanding of abstract ideas: 1) by breaking down complex abstract ideas into simpler viewpoints, it becomes easier for humans to understand (Cowell et al., 2019; Rips et al., 2012); 2) showing the connections between the complex ideas and other ideas can also improve human's understanding on these complex ideas (Huang, 2020; Hayes & Kraemer, 2017; Khatin-Zadeh & Farsani, 2022).

Inspired by the psychological findings above, we propose `GraphEval`, a lightweight graph-based LLM framework for idea evaluation, which breaks down complex ideas into simple and comprehensible viewpoints, and bridges different viewpoints into a viewpoint-graph. Specifically, we first deconstruct complex and difficult-to-understand ideas into simple viewpoints using a prompt-based approach with a (small) LLM. Then, we treat each viewpoint as a node and construct edges through LLM-based relation extraction and/or BERT similarity scores. Finally, we create a viewpoint-graph by joining the viewpoints and edges across different ideas. Based on this, we propose two lightweight graph-based methods for idea evaluations: 1) GraphEval-LP: It is a training-free framework based on graph label propagation algorithm. It operates by transferring quality labels from labeled nodes to unlabeled nodes through weighted edges, ultimately predicting the final evaluation of an idea based on the labels of the viewpoint-subgraph associated with it. 2) GraphEval-GNN: It is a deep learning framework based on Graph Neural Networks (GNN) that requires minimal training. It models idea evaluations as a graph prediction problem using GNNs, obtaining evaluation results by predicting the attributes or classifications of viewpoint-subgraphs. Moreover, in order to objectively assess the novelty of ideas, we have also added a plagiarism detection mechanism to the GraphEval-GNN.

Specifically, we have incorporated temporal information into the features of the viewpoint-nodes and deliberately constructed plagiarized ideas along with their negative evaluation labels as negative samples in the GNN training process. By doing so, we enable the `GraphEval` to learn to give a lower evaluation to those ideas that are plagiarisms of previous research.

In summary, our main contributions are as follows:

- To the best of our knowledge, we are the first to investigate LLM-based idea evaluation from a graph perspective, offering new insights into graph-enhanced LLM research.
- We propose a lightweight graph-based LLM framework called `GraphEval` for idea evaluations, which includes GraphEval-LP and GraphEval-GNN methods. It breaks down the complex ideas into simple viewpoints, connects the viewpoints into a viewpoint-graph, and models the idea evaluation as a node-level prediction task on the viewpoint-graph.
- Extensive experiments on two datasets have demonstrated that `GraphEval` can achieve at least a 14% improvement in F1 score with low computation cost and API cost compared with other baselines. Additionally, `GraphEval` can effectively detect plagiarized ideas and provide a fair evaluation.

## 2 RELATED WORKS

**Automatic Idea Evaluation.** The rapid growth of research idea production and intricate knowledge specialization challenge conventional scientific feedback mechanisms (Liang et al., 2024), prompting researchers to explore AI for automated idea evaluation to accelerate the academic innovation cycle. For example, Sun & Li (2021); Li et al. (2019) investigated the use of CNNs for evaluating academic innovation and design, while Siemon (2023); Bell et al. (2024) analyzed the automated idea evaluation process from a Human-Computer Interaction perspective. In addition, numerous studies employed fine-tuned lightweight language models (e.g., BERT (Devlin, 2018)) to evaluate complex texts, such as dialogues (Thida, 2021), tweets (Pota et al., 2021), and the novelty of ideas (Just et al., 2024; Yuan et al., 2022). However, most of these methods require extensive training on large-scale data and face limitations in generalizability (Sun & Li, 2021; Just et al., 2024). Conversely, recent studies have sought to leverage the domain knowledge and logical capabilities of LLMs to create idea evaluators (Ubonsiri, 2024; Baek et al., 2024; Du et al., 2024; Lin et al., 2023a). Du et al. (2024) proposed using a prompt-based approach to allow LLMs to act as reviewers and meta-reviewers in order to assess the level of papers/ideas based on different evaluation criteria. Xu et al. (2024) utilized representations from specific layers of LLMs for evaluation, Shankar et al. (2024) aligned LLM evaluators with human preferences through feedback, and Lu et al. (2024) enhanced the decision-making ability of LLM-based evaluators via self-reflection, few-shot learning, and response integration. Furthermore, Si et al. (2024) measured the consistency gap between LLMs and human expert reviews. However, when faced with the inherently complex semantic information of research ideas (Baek et al., 2024) and the subjectivity of the evaluation task (Si et al., 2024), the decision-making consistency between LLMs and human reviewers remains limited, often leading to LLMs struggling to provide high-quality feedback (Si et al., 2024; Liang et al., 2024; Lu et al., 2024). Recently, some research works have been evaluating long-form texts, such as biographies of people (Min et al., 2023) and complex mathematical reasoning texts (Lightman et al., 2023). These studies divide the long text into multiple subsets and evaluate each of them. Inspired by these works, we decompose the obscure ideas into simple, understandable viewpoint nodes using LLMs, and further evaluate the idea based on graph algorithms.

**Graph for LLMs.** The use of graphs in conjunction with LLMs is an emerging research area, with several established directions. These include integrating LLMs with path selection mechanisms to learn unified graph representations (Shang et al., 2024); constructing graph-based text indexes using LLMs to answer questions over private text corpora (Edge et al., 2024); and utilizing LLMs for knowledge graph creation (Yang et al., 2024; Zhu et al., 2024; Carta et al., 2023; Trajanoska et al., 2023) and completion (Yao et al., 2023b). In addition, Zhang et al. (2024b) proposed the NLGift benchmark, which focuses on the evaluation of LLM graph reasoning generalization; Perozzi et al. (2024) introduced GraphToken, which can explicitly represent structured data for LLMs; Shi et al. (2024) introduced a novel recommender that synergizes LLMs and KGs to enhance recommendations and provide interpretable results. In terms of open-source software, various graph databases are supported by both the LangChain (LangChain, 2024) and LlamaIndex (LlamaIndex, 2024) libraries.

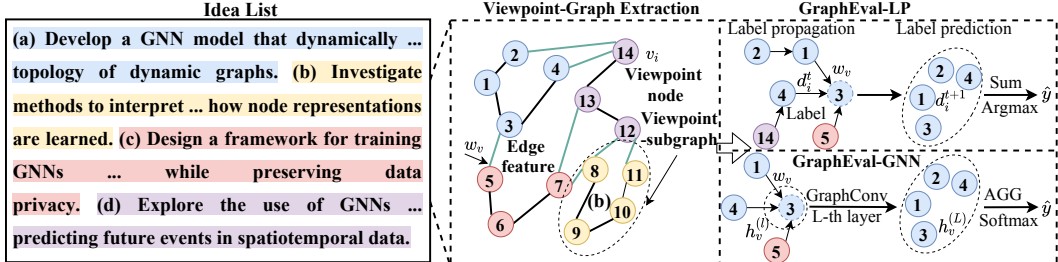

Figure 3: **Overview of `GraphEval` methodology.** `GraphEval` first transforms the ideas into a viewpoint-graph via Viewpoint-Graph Extraction, which contains multiple viewpoint-subgraphs, viewpoint-nodes, and edges between viewpoint-nodes. Then two lightweight `GraphEval` implementations named GraphEval-LP and GraphEval-GNN are employed to evaluate the ideas. Note that AGG denotes the acronym for aggregation function.

However, leveraging LLMs to extract diverse viewpoints embedded in research ideas, structuring them as graphs, and using these for idea evaluation remains a direction that warrants further exploration.

# 3 VIEWPOINT-GRAPH EXTRACTION: A GRAPH STRUCTURE FOR DIVERSE RESEARCH VIEWPOINTS AND THEIR RELATIONSHIPS

**Problem Setup** We consider a predefined set of quality labels $S_{label}$ for evaluating research ideas (e.g., categorical values [Reject, Accept (Poster), Accept (Oral)]). Given a set of ideas $[D_0, D_1, \ldots, D_n]$, only a subset of these ideas has known quality labels during training, and our objective is to predict the quality labels for the remaining ideas at test time.

**Framework Overview** Figure 3 provides an overview of the proposed `GraphEval` framework. The key insight of our approach is that by leveraging the summarization capabilities of LLMs (Jin et al., 2024; Ghosh et al., 2024), we can extract a **viewpoint-subgraph** from each idea's text, which serves as a high-granularity representation that captures diverse viewpoints and the semantic relationships between them. Additionally, we connect multiple viewpoint-subgraphs to construct a larger graph structure, the **viewpoint-graph**, which acts as an extensible database, encompassing existing research viewpoints and their intricate interrelations. This allows us to apply label propagation or GNN algorithms to evaluate ideas in the test set, using only the quality information from the training set ideas.

**Viewpoint Extraction through Prompted LLMs** The key challenges in LLM-based research idea evaluations are twofold: (1) Research ideas inherently encapsulate complex semantic information (Baek et al., 2024; Si et al., 2024), as a single idea often contains multiple distinct viewpoints rooted in different concepts, interconnected through intricate logical relationships that collectively define the idea. (2) Idea evaluation is fundamentally subjective (Si et al., 2024), which presents a significant challenge for LLMs' comprehension and reasoning abilities (Santu et al., 2024; Liang et al., 2024), often resulting in severe biases and a lack of alignment with human evaluations (Lu et al., 2024).

To address these challenges, we utilize LLMs to extract fine-grained components from research ideas, which we refer to as **viewpoints**. A viewpoint can be an idea, argument, or fact embedded within the research content. These viewpoints are semantically independent, evaluable units that are made as granular as possible to ensure they cannot be further decomposed. For a given research idea $D_i$, we employ a prompted LLM $L_p$ to extract a list of viewpoints: $[v_0^i, v_1^i, \ldots, v_k^i] = L_p(D_i)$. A simple example of viewpoint extraction is illustrated in Appendix A.

By extracting viewpoints, we decompose semantically intricate research ideas into fine-grained and semantically independent units. In this process, we utilize prompted LLMs to extract elements as an objective and straightforward NLP task (Manning, 1999; Rush, 2015), relying solely on the models' summarization and abstraction capabilities (Jin et al., 2024; Kurisinkel & Chen, 2023). This approach typically induces fewer biases compared to subjective judgment tasks that necessitate deeper comprehension and reasoning of complex text (Zhang et al., 2023; Zheng et al., 2023). Additionally, it allows us to leverage smaller LLMs (e.g., those with 7 billion parameters) to implement the GraphEval framework, resulting in significant resource savings.

Table 1: **LLM-based relation extraction yields relatively few relevant relationships from the viewpoints, resulting in viewpoint-subgraphs with overly sparse edges.** We conduct LLM-based viewpoint and relation extraction on 300 research ideas, quantifying the average length of each idea and extracted viewpoints, the average number of viewpoints extracted per idea, the average edge number and edge density of each viewpoint-subgraph, with text length measured in word count. The LLM used is Mistral (7B) Instruct v0.3 (Jiang et al., 2023).

| Avg. Idea Len. | Avg. Viewpoint Num. | Avg. Viewpoint Len. | Avg. Edge Num. | Avg. Edge Density |
|---|---|---|---|---|
| 174.60 | 8.83 | 20.19 | 3.71 | 10.73% |

**Viewpoint-subgraph Construction through Prompted LLMs** To identify the semantic relationships between viewpoints extracted from an idea, we utilize prompted LLMs for relation extraction (Wei et al., 2024; Jinensibieke et al., 2024). Specifically, we treat each viewpoint as a node in a graph, referred to as a viewpoint-node. We then input the viewpoint list into the prompted LLM, instructing it to extract semantically related viewpoint pairs. These pairs are subsequently considered as edges connecting the corresponding viewpoint-nodes. We refer to the graph constructed from a research idea as a viewpoint-subgraph.

To validate the feasibility of using prompted LLMs for relation extraction, we collect 300 submissions to the ICLR conferences between 2021 and 2023, treating the abstracts of these academic papers as representations of research ideas (Si et al., 2024; Baek et al., 2024). We perform LLM-based viewpoint and relation extraction on these research ideas. As shown in Table 1, the LLM-based relation extraction yields relatively few relevant relationships from the viewpoints, resulting in viewpoint-subgraphs with overly sparse edges. This leads to an excess of isolated viewpoint-nodes in each viewpoint-subgraph and a deficiency in the inherent relational information. Additionally, the LLM-based relation extraction incurs extra resource costs.

To address these issues, we propose a method for relation extraction based on embedding similarity.

**Viewpoint-subgraph Construction through BERT-based Encoder** To automatically identify logical relationships between viewpoints, we use a BERT-based encoder $E$ to obtain embeddings of equal dimensions $e$ for each viewpoint-node $v$ (Qasim et al., 2022; Lin et al., 2023b): $[e_1, e_2, \ldots, e_n] = E([v_1, v_2, \ldots, v_n])$. Then, we compute the cosine similarity $s$ between their embeddings: $s(e_i, e_j) = \frac{e_i \cdot e_j}{\|e_i\|\|e_j\|}$ (Izacard et al., 2021; Li et al., 2023). Each viewpoint-node is connected to the top-$k$ nodes with the highest embedding cosine similarity using weighted undirected edges, with the edge weights set to the cosine similarity (Harnoune et al., 2021). This way, we construct the viewpoint-subgraph, which serves as a high-granularity representation of the research idea. Additionally, by controlling the value of $k$, we can regulate the edge density, allowing for the construction of viewpoint-subgraphs that are more suited to specific downstream tasks.

**Viewpoint-graph Construction through Connecting Viewpoint-subgraphs** After transforming the ideas in both the training set and test set into viewpoint-subgraphs, we connect them to construct a larger graph. Specifically, similar to the construction of viewpoint-subgraphs, for each viewpoint-node, we connect it to the top-$m$ nodes from different subgraphs with the highest embedding cosine similarity using undirected weighted edges. We refer to the graph constructed as the viewpoint-graph, which integrates the diverse viewpoints of different research ideas and the interrelations between them. The viewpoint-graph $G$ can be represented by a node list and an edge list:

$$G = \{[(v_0, e_0), ..., (v_n, e_n)], [(v_{k_0}, v_{k_1}, w_{k_0 k_1}), ..., (v_{k_{mn}}, v_{k_{mn+1}}, w_{k_{mn}k_{mn+1}})]\} \quad (1)$$

Notably, the viewpoint-graph is scalable, allowing new viewpoint-subgraphs to be integrated in linear time, providing a theoretical foundation for its expansion as new ideas are generated.

## 4 GRAGHEVAL-LP: A SIMPLIFIED AND LIGHTWEIGHT IMPLEMENTATION

After obtaining the viewpoint-graph $G$, we would like to validate its efficacy by first applying a simple and lightweight algorithm, label propagation (Raghavan et al., 2007; Zhang et al., 2017), to evaluate the ideas in the test set. Our results in Section 7 show that this simple algorithm is already very effective with idea evaluations. We refer to this evaluation framework as **GraphEval-LP**.

**Initialization and Regularization** For each viewpoint-node $v_i$ in $G$, we maintain a vector $d_i$, where each dimension corresponds to a quality label in $S_{label}$. Thus, the dimensionality of $d_i$ is given by

$|d_i| = |S_{label}|$. For viewpoint-nodes extracted from ideas in the training set, we assign a value of 1 to the dimension corresponding to the idea's label, while all other dimensions are set to 0. In contrast, viewpoint-nodes extracted from the test set are initialized as zero vectors. Additionally, we regularize the edge weights $w_{ij}$ in $G$ to ensure that the sum of the weights of all edges connected to any given viewpoint-node $v_i$ equals 1, i.e., $\sum_{j \in N(i)} w_{ij} = 1$, where $N(i)$ represents the set of neighbors of $v_i$.

**Label Propagation** We perform multiple iterations of label propagation on graph $G$ until the labels no longer change. Specifically, in each iteration, each node updates its vector by adding the weighted vectors of its neighboring nodes:

$$d_i^{(t+1)} = \frac{1}{Z_i}(d_i^{(t)} + \sum_{j \in N(i)} w_{ij} d_j^{(t)}) \tag{2}$$

Where $d_i^{(t)}$ is the vector of node $v_i$ at iteration $t$, and $Z_i$ is a normalization factor that ensures the updated vector is properly scaled.

**Label Prediction** After completing label propagation, we sum the vectors of the viewpoint-nodes corresponding to each idea in the test set. The predicted label $\hat{y}$ is then determined by selecting the dimension with the highest value in the summed vector, i.e., $\hat{y} = \arg\max_j \left( \sum_{i=1}^{k} d_i \right)_j$, where $j$ indexes the dimensions of the vector and $k$ means the number of viewpoints for a given research idea.

## 5 GRAGHEVAL-GNN: A GRAPH NEURAL NETWORK-BASED SOLUTION

Although label propagation is effective, it does not learn how to properly propagate evaluation scores from known nodes to unknown nodes. Therefore, we further propose a learning-based approach, **GraphEval-GNN**, which is trained to predict the evaluation scores for a viewpoint-node.

**Method Overview.** As shown in Figure 3, GraphEval-GNN models viewpoints as viewpoint-nodes, while the relationships between viewpoints are represented by edge features. We apply GNN to embed the node and edge features and use them for training and testing.

**Initialize node/edge features.** As illustrated in Sec. 3, we initialize the viewpoint-node features $\mathbf{h}_v$ by converting viewpoints into embeddings using BERT. Since the relationships between viewpoint-nodes encompassing the similarity relations obtained from BERT, we initialize the edge features $w_v$ using this relational attribute.

**Predict via a weighted GNN.** We implement the predictive model $f_\phi$ over viewpoint-nodes using a weighted GNN, as shown in Figure 3. The objective of the GNN is to learn expressive node embeddings $\mathbf{h}_v$ through an iterative weighted aggregation of the local network neighborhoods. The $l$-th iteration of the GraphConv$(\cdot)$, or the node embeddings update of the $l$-th layer, is represented as:

$$\mathbf{h}_v^{(l)} = \mathbf{U}^{(l)} \text{CONCAT}\Big( \text{MEAN}\Big( \{\text{RELU}(\mathbf{w_v} \mathbf{W}^{(l)} \mathbf{h}_q^{(l-1)}), q \in N(v)\}\Big), \mathbf{h}_v^{(l-1)}\Big), \tag{3}$$

where $\mathbf{h}_v^{(l)}$ is the node embedding after $l$ iterations, $\mathbf{h}^{(0)}$ have been initialized as explained above, $N(v)$ denotes the direct neighbors of $v$, and $\mathbf{U}^{(l)}, \mathbf{W}^{(l)}$ are learnable parameters.

Since the evaluation of an idea is determined by all the viewpoints extracted from it, we further model the LLM evaluation problem as a sub-graph prediction problem and aggregate all the node embeddings into a subgraph embedding. Moreover, as introduced in Figure 2, we consider MEAN Pooling and Max Pooling simultaneously to extract global and local information of the idea. Specifically, the sub-graph probability distribution $\hat{y}_{D_i}$ of idea $D_i$ can be made through GraphPred$(\cdot)$ in the form of:

$$\hat{y}_{D_i} = \text{SOFTMAX}\left( \text{MLP}\left( \text{CONCAT}\left( \text{MEAN}\left\{\mathbf{h}_v^{(l)} : v \in L_p(D_i)\right\}, \text{MAX}\left\{\mathbf{h}_v^{(l)} : v \in L_p(D_i)\right\}\right)\right)\right), \tag{4}$$

We have summarized the detailed training process of `GraphEval` in Algorithm 1. In addition, in the testing of `GraphEval`, we choose the category with the highest output probability as the result of the LLM evaluation.

---

**Algorithm 1** Training of `GraphEval`

---

**Require:** Dataset $\mathcal{D}_{\text{train}} = \{(\mathbf{x}, y)\}$. A weighted GNN $f_\phi$. Edge weights $\mathbf{w}_v$. Number of GNN layers $L$.

1: Initialize the embeddings of viewpoint node, $h_v^{(0)}$, using BERT.
2: **for** each iteration $i$ **do**
3: $\quad$ $N \leftarrow \text{SampleMiniEdgeBatch}(\mathcal{D}_{\text{train}})$
4: $\quad$ Mask the viewpoint-subgraphs in $\mathcal{D}_{\text{train}}$ that are in $N$, and obtain the labels of the viewpoint-subgraphs in $T_n^{(i)}$
5: $\quad$ **for** $l = 1$ to $L$ **do**
6: $\quad\quad$ $\mathbf{h}_v^{(l)} \leftarrow \text{GraphConv}(\mathbf{h}_v^{(l)}, \mathbf{w}_v)$ with $f_\phi$
7: $\quad$ Backward $\left( \text{Criterion} \left( \hat{y}_{D_i}, \{y_j\}_{j \in T_n^{(i)}} \in N \right) \right)$

---

**`GraphEval` for idea novelty assessment.** Assessing the novelty of ideas is crucial, as plagiarized or derivative ideas can sometimes mislead LLMs into giving them higher evaluation scores. As a concrete example, if the same idea is being evaluated by an LLM twice, LLM will always assign the same evaluation score, since it does not take the novelty aspect into account when evaluating ideas.

To address this issue, we enforce our GraphEval-GNN to learn that ideas and viewpoints appearing later in time and exhibiting high similarity to earlier ones should be evaluated with lower scores. Specifically, our approach focuses on two key aspects. First, we incorporate temporal features into the viewpoint representations, enabling the model to capture the chronological sequence of viewpoints. Second, we artificially generate duplicated ideas and viewpoints that are direct combinations of existing viewpoints in the viewpoint-graph, label them with lower evaluation scores as negative samples, and include them in the GNN training process.

## 6 EXPERIMENTAL SETUP

**Task.** Following the works of Si et al. (2024); Baek et al. (2024), we treat the abstract of an academic paper as a representation for the research idea, since it typically offers a concise summary of the research problem, the scientific methods employed, the experimental design, and the key contributions. Specifically, we provide each method with the abstracts and titles of the academic papers, tasking them with evaluating the review decision: Reject, Accept (Poster), Accept (Oral), or Accept (Spotlight).

**Datasets.** We employ two datasets to thoroughly evaluate the proposed GraphEval framework:

- ICLR Papers: We collect abstracts and review decisions from paper submissions to the ICLR conferences between 2021 and 2023. From this, we randomly select 300 papers as the training set for learning-based methods and 50 papers as the test set.
- AI Researcher Dataset: We use the dataset collected by Si et al. (2024) in AI Researcher as an additional test set, which contains academic papers focusing on the domain of "novel prompting methods." Note that due to the scarcity of Accept (Oral) and Accept (Spotlight) labels in this dataset, we combine them into a single label, thereby transforming the task into a three-class classification problem.

The details of the datasets can be found in Appendix C.

**Baselines.** To gain a comprehensive understanding of the performance of our proposed framework in evaluating research ideas, we have adopted several baselines:

- **Prompted LLM**: We provide several criteria for assessing research ideas in the prompt. Additionally, we present specific standards for the four review decisions and include one example for each as few-shot examples for in-context learning (Brown, 2020). Moreover, we include the label distribution of the dataset to help the LLMs understand the frequency of each review decision.
- **CoT prompt**: Drawing inspiration from Wei et al. (2022), we modify the prompt used for prompted LLM to adopt a CoT format, guiding it to complete the idea evaluation step by step.
- **CoT-SC**: Self-consistency with CoT (CoT-SC) is an ensemble approach that samples $k = 5$ i.i.d. CoT, then returns the most frequent output (Wang et al., 2022).

- **ToT prompt**: Tree of Thoughts (ToT) is an extension of CoT (Yao et al., 2024). Similar to CoT, we divide the evaluation process into multiple steps. At each step, we sample $branch = 5$ i.i.d. CoTs, and pass the most frequent output as the intermediate result to the next step.
- **Research Agent**: We adopt the idea evaluation method from Research Agent (Baek et al., 2024) as one of our baselines, where the research problem, scientific method, and experiment design of an idea are each scored based on five criteria. Building on this, we further introduce a final decision step that synthesizes the above evaluation results to provide a comprehensive review decision.
- **Fine-tuned BERT**: In addition to LLM-based methods, we fine-tune a DistilBERT model (Sanh et al., 2019) using collected paper abstracts and review decisions as a baseline to validate the competitiveness of our approach compared to learning-based methods.

For all LLM-based baselines (except Fine-tuned BERT), we use two LLMs of different sizes: Mistral (7B) Instruct v0.3 (Jiang et al., 2023) and Qwen 2 Instruct (72B) (qwe, 2024). All prompts used in the methods can be found in the Appendix B.

**Evaluation Metrics.** To comprehensively evaluate the consistency between the idea evaluation methods and human reviewers, we calculate the **accuracy**, **macro precision**, **macro recall**, and **macro F1 score** for each method. Additionally, we record the average **token cost** per evaluation as a measure of resource consumption. Note that for Mistral (7B) Instruct v0.3, the API cost is \$0.20 per 1M tokens, and for Qwen 2 Instruct (72B), the API cost is \$0.90 per 1M tokens.[1] We calculate the average cost per evaluation for each method according to these pricing standards. To intuitively illustrate the resource consumption of each method, we normalize the average costs by setting the highest-cost method to 1, which we refer to as **normed cost**. A smaller normed cost indicates lower resource expenditure.

**Implementation Details.** During the training phase, we configured the graph neural network as a two-layer weighted GNN with a hidden dimension of 64. The batch size is set to 64, and the maximum number of training epochs is limited to 1000. We employ the Adam optimizer (Diederik, 2014) for training and gradually reduce the learning rate from 1e-3 to 0 using a LambdaLR scheduler. Our proposed method is implemented using PyTorch[2] and PyTorch Geometric (PyG)[3], with all experiments conducted on a single NVIDIA A100 Tensor Core GPU. For the LLMs, we utilize API calls from Together AI[4] to obtain responses. Additionally, the average GPU memory usage of GraphEval-GNN for the two tasks is 372MB, whereas Fine-tuned BERT utilizes 4.84 GB on average.

## 7 EXPERIMENT RESULTS

### 7.1 COMPARISON WITH EXISTING BASELINES.

We report the performance of our methods and baselines in Tables 2 and 3.

(1) Across all datasets, GraphEval-GNN significantly outperforms all baselines: for the ICLR Papers dataset, it achieves a 10%-72% accuracy advantage and an 18%-42% macro F1 score advantage; for the AI Researcher dataset, it achieves a 13%-53% accuracy advantage and a 14%-48% macro F1 score advantage. Moreover, its normed cost on both datasets demonstrates its utilization of resources comparable to the minimum expenditure level. This indicates that by leveraging a smaller LLM (7B parameters) to convert semantically complex research ideas into more granular viewpoint-graph, and utilizing GNN algorithms to extract global and local information, we achieve precise evaluation of research ideas.

(2) Regarding the prompt-based baselines, they generally achieve lower accuracy and macro F1 scores. Our observations indicate that all these methods tend to overestimate the quality of ideas, with very few ideas being rejected. This aligns with findings in previous works (Lu et al., 2024; Si et al., 2024). Furthermore, we find that using larger-scale LLMs does not consistently improve evaluation performance; rather, it often leads to a decline: In experiments, the 72B model tends to provide consistent or similar decisions and overall scores for different ideas. This suggests that LLMs exhibit

---

[1]The API pricing referenced here is based on the rates provided by https://www.together.ai/pricing.

[2]https://pytorch.org/

[3]https://pytorch-geometric.readthedocs.io/en/latest/

[4]https://www.together.ai/

Table 2: **GraphEval-GNN consistently outperforms all baselines in the ICLR Papers Dataset while utilizing resources comparable to the minimum expenditure level.** Bold and underlined text denotes the best and second-best results. Specifically, for accuracy, macro precision, macro recall, and macro F1 score, higher values indicate more precise predictions of the labels for research ideas in the test set. Conversely, for the normed cost, lower values represent reduced resource expenditure. Since Fine-tuned BERT is not an LLM-based method, its token cost and normed cost are not calculated.

| Dataset | ICLR Papers | | | | | |
|---|---|---|---|---|---|---|
| Method\Metric | Accuracy | Precision | Recall | F1 Score | Token Cost | Normed Cost |
| **Prompted LLM (7B)** | 16.00% | 4.55% | 16.67% | 7.14% | 1968.22 | **0.06** |
| **Prompted LLM (72B)** | 6.00% | 4.09% | 31.25% | 6.35% | 1735.30 | 0.24 |
| **CoT prompt (7B)** | 16.00% | 5.00% | 16.67% | 7.69% | 2443.28 | 0.07 |
| **CoT prompt (72B)** | 6.00% | 5.36% | 27.08% | 5.05% | 2415.62 | 0.33 |
| **CoT-SC (7B)** | 20.00% | 5.21% | 20.83% | 8.33% | 3121.72 | 0.10 |
| **CoT-SC (72B)** | 4.00% | 1.19% | 25.00% | 2.27% | 3428.14 | 0.47 |
| **ToT prompt (7B)** | 8.00% | 4.95% | 18.75% | 6.47% | 8963.92 | 0.27 |
| **ToT prompt (72B)** | 4.00% | 1.06% | 25.00% | 2.04% | 6211.46 | 0.85 |
| **Research Agent (7B)** | 12.00% | 8.11% | 22.92% | 10.05% | 7909.18 | 0.24 |
| **Research Agent (72B)** | 6.00% | 5.30% | 31.25% | 7.17% | 7278.72 | 1.00 |
| **Fine-tuned BERT** | 66.00% | 27.22% | 28.39% | 26.01% | \ | \ |
| **GraghEval-LP (Ours)** | 70.00% | 37.61% | 32.55% | 32.16% | 2672.95 | 0.08 |
| **GraghEval-GNN (Ours)** | **76.00%** | **56.59%** | **42.63%** | **43.59%** | 2672.95 | 0.08 |

Table 3: **GraphEval-GNN consistently outperforms all baselines in the AI Researcher Dataset while utilizing resources comparable to the minimum expenditure level.** Bold and underlined text denotes the best and second-best results. Since Fine-tuned BERT is not an LLM-based method, its token cost and normed cost are not calculated.

| Dataset | AI Researcher | | | | | |
|---|---|---|---|---|---|---|
| Method\Metric | Accuracy | Precision | Recall | F1 Score | Token Cost | Normed Cost |
| **Prompted LLM (7B)** | 26.98% | 50.40% | 36.44% | 24.75% | 1961.41 | **0.06** |
| **Prompted LLM (72B)** | 30.16% | 52.88% | 41.99% | 28.33% | 1717.57 | 0.23 |
| **CoT prompt (7B)** | 30.16% | 51.44% | 37.09% | 21.18% | 2410.06 | 0.07 |
| **CoT prompt (72B)** | 23.81% | 50.51% | 34.97% | 22.86% | 2263.92 | 0.31 |
| **CoT-SC (7B)** | 31.75% | 26.61% | 41.67% | 26.43% | 2854.44 | 0.09 |
| **CoT-SC (72B)** | 25.40% | 52.36% | 37.75% | 24.37% | 3157.40 | 0.43 |
| **ToT prompt (7B)** | 30.16% | 42.66% | 29.04% | 19.89% | 9829.14 | 0.30 |
| **ToT prompt (72B)** | 25.40% | 51.78% | 39.38% | 22.93% | 6071.98 | 0.83 |
| **Research Agent (7B)** | 27.42% | 19.78% | 38.19% | 24.24% | 7887.44 | 0.24 |
| **Research Agent (72B)** | 20.63% | 14.53% | 32.03% | 18.71% | 7345.06 | 1.00 |
| **Fine-tuned BERT** | 60.00% | 54.44% | 54.44% | 53.33% | \ | \ |
| **GraghEval-LP (Ours)** | 70.47% | 61.11% | 55.56% | 56.97% | 2541.17 | 0.08 |
| **GraghEval-GNN (Ours)** | **73.33%** | **81.67%** | **73.33%** | **67.13%** | 2541.17 | 0.08 |

significant bias when faced with subjective judgment tasks that necessitate a deeper understanding and reasoning of complex text (Zhang et al., 2023; Zheng et al., 2023), irrespective of their model size and capability. On the other hand, the GraphEval framework mitigates bias by transforming the LLM's task into the objective and straightforward task of extracting elements.

(3) Compared to CoT-SC, the ToT prompt and Research Agent, which utilize more complex prompting techniques, do not demonstrate significant advantages. This suggests that prompting techniques have limited efficacy in enhancing the capabilities of LLMs for tasks requiring complex comprehension and reasoning.

(4) Although fine-tuned BERT achieves better results compared to other prompt-based baselines, it still falls short of the performance level of GraphEval. This is due to the construction of the viewpoint-graph, which allows GraphEval-LP and GraphEval-GNN to obtain quality information about viewpoints locally and capture the intricate interrelations among diverse viewpoints globally, thereby leading to improved performance.

(5) GraphEval-LP consistently achieves the second-best results across both datasets, and it does not require training, making it efficient and lightweight. GraphEval-LP effectively demonstrates the

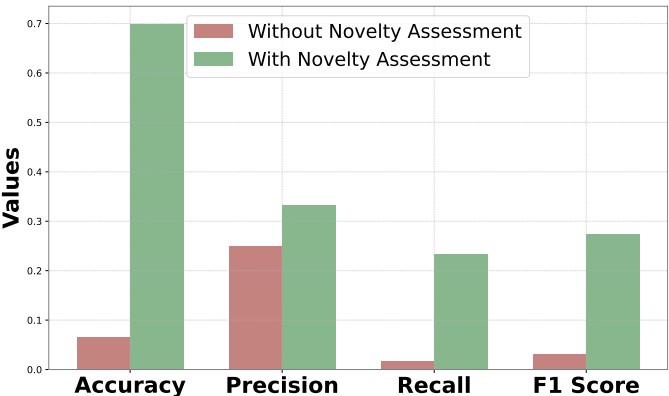

Figure 4: **Novelty assessment can significantly improve the performance of `GraphEval` when detecting plagiarized or derivative ideas.** We compare two variants of `GraphEval` on the ICLR Papers dataset and evaluate their performance on four metrics.

strong utility of the constructed viewpoint-graph for research idea evaluation, owing to the inherent correlations between research ideas, such as shared common viewpoints. These implicit correlations cannot be effectively leveraged by prompt-based methods or fine-tuned language models.

(6) The comparison between GraghEval-LP and GraghEval-GNN demonstrates that: 1) Deep learning can enhance the performance of graphs when applied for LLM evaluation tasks; 2) Although the introduction of GNN has improved performance, it also results in increased computational cost. Therefore, in our paper, we propose these two implementations to provide options for users with different needs.

### 7.2 GRAPHEVAL FOR NOVELTY ASSESSMENT.

To evaluate the effectiveness of novelty assessment on the ICLR Papers dataset, we artificially construct 80 plagiarized ideas for testing. Specifically, we employ three strategies to evenly create these 80 ideas: 1) We directly copy highly-rated ideas from the dataset; 2) We randomly replace some viewpoints in highly-rated ideas with viewpoints from other ideas; 3) We substitute some viewpoints in highly-rated ideas with those of their neighboring nodes based on the similarity of embeddings. Subsequently, we select 10 of the above ideas and construct negative samples using the method mentioned in Sec 5, which are then provided to `GraphEval` for training. We compare the impact of considering Novelty Assessment on the performance of GNN across four metrics, as shown in Figure 4. We can observe that Novelty Assessment can significantly improve the performance of GraphEval when detecting plagiarized or derivative ideas.

## 8 CONCLUSION

In this paper, we propose a novel lightweight graph-based LLM framework, `GraphEval`, for idea evaluation, addressing the complexities and subjectivity inherent in this task. Drawing inspiration from human psychology, we break down complex ideas into simpler viewpoints and model the relationships between them using viewpoint-graphs. Our framework includes two methods: GraphEval-LP, a training-free approach utilizing label propagation, and GraphEval-GNN, a deep learning-based method using Graph Neural Networks. Both methods effectively leverage viewpoint-graphs to predict idea evaluations, while GraphEval-GNN also incorporates a plagiarism detection mechanism that ensures fair and objective assessment of novelty.

Through extensive experiments on two datasets, we demonstrate that `GraphEval` not only achieves a significant improvement in accuracy and macro F1 score compared to multiple baselines but also operates with a low resource expenditure. Our work pioneers the integration of graph-based approaches with LLMs for idea evaluation, providing new insights for enhancing LLM-based evaluations with graph representations.

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

## A    A SIMPLE EXAMPLE OF VIEWPOINT EXTRACTION

Here, we present a simple example of viewpoint extraction in Figure 5. For a given research idea $i$, we employ a prompted LLM $L_p$ to extract a list of viewpoints: $[v_0, v_1, \ldots, v_n] = L_p(i)$.

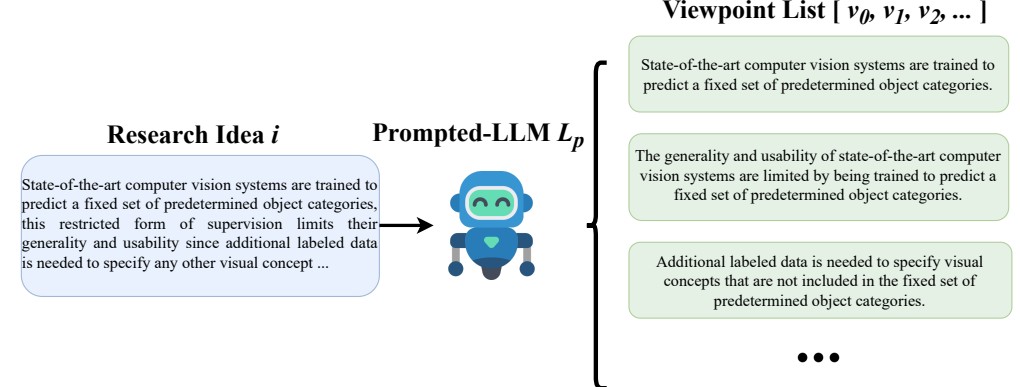

Figure 5: **Example of viewpoint extraction from a research idea.** This figure illustrates how a prompted LLM extracts fine-grained viewpoints from a research idea. Each viewpoint represents an independent, evaluable unit such as an idea, argument, or fact. The viewpoints capture distinct components of the research idea that contribute to its overall understanding.

## B    PROMPT USAGE

Here, we present the prompts used in our method and the baselines.

### B.1    PROMPTS USED IN GRAPHEVAL

We present the prompts used in LLM-based viewpoint extraction and relation extraction in Table 4 and Table 6, respectively.

### B.2    PROMPTS USED IN BASELINES

We present several criteria for evaluating research ideas, along with specific standards for the four review decisions outlined in the prompts used for the baselines. The idea evaluation criteria and decision prompt templates can be found in Table 8 and Table 9.

The prompt used in the prompted LLM is presented in Table 10, while the prompt used in the CoT prompt and CoT-SC is shown in Table 11.

For the ToT prompt, we decompose the problem into eight steps: novelty evaluation step, validity evaluation step, significance evaluation step, rigorousness evaluation step, clarity evaluation step, ethical evaluation step, overall score step, and final discussion step. The prompts used for the novelty evaluation step, validity evaluation step, overall score step, and final discussion step are presented in Tables 12, 13, 14, 15; the prompts for the remaining steps are similar to those displayed in Tables 12 and 13.

Building on the work of Baek et al. (2024), our Research Agent baseline divides the task into four steps: problem validation step, method validation step, experiment validation step, and final decision step, with the corresponding prompts presented in Tables 16, 17, 18, 19.

Table 4: **Viewpoint extraction prompt template**. Here, for brevity and clarity, we have omitted portions of the system's input and the LLM's answer from the one-shot demonstration.

> You are required to act as an AI annotator and extract the Viewpoints embedded in the sentences of the provided academic paper abstract. Below, you will be given an abstract from an academic paper. You need to break it down sentence by sentence and extract the Viewpoints embedded in each sentence. The extracted Viewpoints can be an idea, argument, or fact. Each sentence may contain one or more Viewpoints to be extracted.
> The extracted Viewpoints should be as granular as possible to ensure they cannot be further broken down.
> When extracting Viewpoints from a sentence, pay attention to the context within the abstract. Replace pronouns with the nouns they represent and complete any omitted sentence components to ensure the independence of the Viewpoints is not compromised. This means that each extracted Viewpoint should not contain pronouns whose referents cannot be found within that Viewpoint.
> Below is an example interaction that can serve as a reference for the format and method of extracting Viewpoints:
> System's Input:
> [The Start of Abstract]
> State-of-the-art computer vision systems are trained to predict a fixed set of predetermined object categories. ...
> [The End of Abstract]
> Your Answer:
> [Sentence 1]
> State-of-the-art computer vision systems are trained to predict a fixed set of predetermined object categories.
> [Extracted Viewpoints in Sentence 1]
> [State-of-the-art computer vision systems are trained to predict a fixed set of predetermined object categories.]
> [Sentence 2]
> ...

Table 5: **Details of the Datasets**. We present the data sizes and label distributions for the datasets used in our experiments. For the AI Researcher Dataset, due to the scarcity of Accept (Oral) and Accept (Spotlight) labels, we have combined them into a single label.

| Dataset | Data Size | Reject | Poster | Oral | Spotlight |
|---------|-----------|--------|--------|------|-----------|
| ICLR Papers (Training) | 300 | 55% | 25% | 10% | 10% |
| ICLR Papers (Test) | 50 | 64% | 24% | 8% | 4% |
| AI Researcher Dataset | 66 | 53.03% | 27.27% | 19.70% | |

## C  DETAILS OF THE DATASETS

In this section, we present the details of the ICLR Papers and AI Researcher Dataset used in our experiments, as shown in Table 5.

Specifically, we control the label distribution of the training set in the ICLR Papers by increasing the representation of Accept (Oral) and Accept (Spotlight) papers. This adjustment enables learning-based methods to effectively capture features of less-represented samples under long-tail distribution conditions.

For the AI Researcher Dataset (Si et al., 2024), due to the scarcity of Accept (Oral) and Accept (Spotlight) labels, we have combined them into a single label, thus transforming the task into a three-class classification problem. Additionally, given the limited data volume in this dataset, we record the performance metrics of the methods across the entire dataset when testing prompt-based methods to obtain a more comprehensive evaluation. For testing other methods, we split the dataset into training and testing sets in an 85%:15% ratio and conduct multiple experiments to average the results, thereby reducing bias.

Table 6: **Relation extraction prompt template**. Here, for brevity and clarity, we have omitted portions of the system's input and the LLM's answer from the one-shot demonstration.

You are required to act as an AI annotator. You will be provided with an abstract from an academic paper along with a set of extracted Viewpoints. These Viewpoints can represent an idea, argument, or fact.

Your task is to identify pairs of related Viewpoints and provide a suitable logical connector to describe the relationship between each selected pair. Then, you need to indicate whether the logical relationship you selected belongs to the "supporting relationship" or "opposing relationship" category. The "supporting relationship" can describe logical connections such as continuation, cause-effect, exemplification, etc., while the "opposing relationship" is used to describe contrast, contradiction, etc.

The format of a Viewpoint Pair is as follows: {[Viewpoint1], [logical connector], ["supporting" or "opposing"], [Viewpoint2]}

You need to refer to the given academic paper abstract to determine the relevance between Viewpoints and the appropriate logical relationship for related Viewpoints based on the context. You need to list all the Viewpoint Pairs you find.

Below is an example interaction that can serve as a reference for the format and method of constructing Viewpoint Pairs:

System's Input:

[The Start of Abstract]

State-of-the-art computer vision systems are trained to predict a fixed set of predetermined object categories. ...

[The End of Abstract]

[The Start of Extracted Viewpoints]

[State-of-the-art computer vision systems are trained to predict a fixed set of predetermined object categories.]

...

[The End of Extracted Viewpoints]

Your Answer:

[The Start of Viewpoint Pairs]

{[State-of-the-art computer vision systems are trained to predict a fixed set of predetermined object categories.], [however], [opposing], [The generality and usability of state-of-the-art computer vision systems are limited by being trained to predict a fixed set of predetermined object categories.]}

...

[The End of Viewpoint Pairs]

Table 7: **Hyperparameter Settings for Experiments**. This table lists the hyperparameters, their descriptions, and the values used during our experiments.

| Parameter Name | Description | Value |
|---|---|---|
| temperature t | The temperature coefficient set when calling the LLM | 0.1 |
| intra graph degree k | Degree of each view-node in Sub-viewpoint-graph construction through embedding similarity | 5 |
| inter graph degree m | Number of view-nodes each node connects to in the Viewpoint-graph construction, from a different sub-graph | 10 |
| max_iters | Number of iterations in Label Propagation for GraphEval-LP | 5 (ICLR Papers) 2 (AI Researcher) |

# D HYPERPARAMETER CONFIGURATION

The hyperparameter settings used in our experiments are presented in Table 7.

Table 8: **Idea evaluation criteria prompt template.** We outline several criteria for assessing research ideas in the prompts used for the baselines.

| Criteria | Texts |
|---|---|
| Novelty | Does it introduce a new problem or perspective that has not been explored before? Does it introduce new techniques or represent a significant advancement compared to existing methods? How does it align with or diverge from current research trends? |
| Validity | Does it include solid theoretical foundations, robust algorithms, and detailed methodologies to address the research problem? Are the underlying principles well-defined and logically consistent? |
| Significance | Consider its potential contribution and impact on the research community in its specific domain and beyond. How does it compare to existing works in terms of impact? |
| Rigorousness | Are the research design and methods clearly described and justified? Is the methodology robust and appropriate for addressing the research questions? Are the results well-analyzed and interpreted? Do the findings support the claims made in the paper? |
| Clarity | How well do the title and abstract summarize the paper? Are they clear, concise, and informative? Does the paper effectively convey its significance and main contributions? Are the title and abstract well-aligned with each other and accurately represent the core idea and content of the paper? Is the content well-structured and easy to follow? |
| Ethical Considerations | Does it adhere to ethical guidelines and responsible research practices? Are potential negative consequences or biases addressed? |

Table 9: **Idea evaluation decision prompt template.** We present specific standards for the four review decisions in the prompts used for the baselines.

| Decision | Texts |
|---|---|
| Reject | Papers in this category lack sufficient novelty, contain fundamental flaws in methodology, or fail to present a significant contribution to the field. For example, a paper that proposes a minor tweak to existing methods without offering substantial improvement may fall under this category. |
| Accept (Poster) | These papers offer incremental contributions, demonstrate solid theoretical or experimental work, and may be of interest to a niche audience. They have clear and understandable results but may not present breakthroughs. |
| Accept (Oral) | Papers in this category present more significant contributions to the field, with clear and convincing evidence of effectiveness. The methodology is robust, and the findings are impactful. These papers are well-executed and can be of interest to a broader audience. |
| Accept (Spotlight) | Papers that represent groundbreaking work or a major advancement in the field, offering novel insights or techniques with broad applicability and significant potential impact. These papers stand out in terms of both innovation and technical quality. |

# E    GENERALIZATION EXPERIMENT

## E.1    GENERALIZATION ON LONG FORM TEXT EVALUATION TASK

To validate GraphEval's capability in text evaluation forms beyond research ideas, we conducted experiments on a long form text evaluation task (Min et al., 2023). Specifically, we used human-annotated data from the FActScore dataset, where each entry contains "atomic facts" about celebrities generated by LLMs, along with assessments from human annotators on whether these "atomic facts" were supported by the materials provided to the annotators. Based on the "atomic facts" and human annotations from the training set, our method needed to predict the labels of "atomic facts" in the test set that were partitioned off. We selected topics such as Ramesses IV, Lanny Flaherty, and Florencia

Table 10: **Prompted LLM prompt template**. We provide several criteria for assessing research ideas in the prompt. Additionally, we present specific standards for the four review decisions and include one example for each as few-shot examples for in-context learning. Moreover, we include the label distribution of the dataset to help the LLMs understand the frequency of each review decision.

[System Prompt]
You are an AI researcher who is reviewing a paper's title and abstract that was submitted to a prestigious ML venue. Be critical and cautious in your decision.
If a paper's title and abstract are bad or you are unsure, give it bad scores and reject it!
[Instruction]
Please evaluate the paper draft based on the following six dimensions:
{idea evaluation criteria prompt template}
You will classify the paper into one of the following four categories based on the evaluation:
{idea evaluation decision prompt template}
**Note:** The approximate distribution of decisions for papers at this ML venue is as follows:
{label distribution of the dataset}. Please take this decision distribution into account and make your judgment carefully.
[Examples for Evaluation Standards]
{one example per decision}
[Input]
Here is the paper draft to evaluate: Title – {title}; Abstract – {abstract};
[Output]
You only need to give an overall score (0-100) and select a review decision. No detailed analysis is required.
The output format should follow these rules:
Overall Score (0-100)= {score}
{one decision from "Reject", "Accept (Poster)", "Accept (Oral)", "Accept (Spotlight)"}
An example of the output:
Overall Score (0-100)= 82
Reject

Bertotti, and divided the training, validation, and test sets in a 7:1:2 ratio. We compared GraphEval and some applicable baselines on this dataset in Table 20. The experimental results in the table verify that our approach performs well on the long form text evaluation task, demonstrating good adaptability to various tasks.

### E.2  GENERALIZATION ABILITY ACROSS DIFFERENT TIMES

To explore the temporal generalization performance of GraphEval on the dataset, we selected papers from before 2022 in the ICLR Papers dataset as the training and validation sets, and papers from 2023 as the test set. We compared the performance of GraphEval with other classic baselines in Table 21. The results in the table validate GraphEval's temporal generalization ability in the task of idea evaluation.

## F  ADDITIONAL ABLATION STUDY

### F.1  EFFECTS OF VARIOUS LIGHTWEIGHT GRAPH NEURAL NETWORK ARCHITECTURES

To compare the impact of different lightweight GNN architectures on the performance of GraphEval, we selected two classic lightweight GNN frameworks, SGC (Wu et al., 2019) and LightGCN (He et al., 2020), to replace the current heterogeneous graph structure in GraphEval. We named these two baselines GraphEval-SGC and GraphEval-LightGCN, respectively. We compared these baselines with GraphEval-GNN on the ICLR Papers dataset, as shown in 22. We observed that the performance of the lightweight frameworks was inferior to that of GraphEval-GNN, which is due to their sacrifice of individualized node information in order to optimize memory usage and speed.

### F.2 COMPARATIVE IMPACT OF ALTERNATIVE RELATION EXTRACTION METHODS

We proposed a hybrid relation extraction method named Hybrid to compare with our fully similarity-based approach, GraphEval. Specifically, the hybrid method uses Prompted LLMs mentioned in Section 3 to connect nodes within viewpoint-subgraphs, while the edges between viewpoint-subgraphs are still based on similarity. The results of the two relation extraction methods on the ICLR Papers dataset are presented in Table 23, showing that GraphEval-GNN performs better than Hybrid. This might be due to the difficulty of ensuring adequate edge density when connecting nodes within viewpoint-subgraphs using Prompted LLMs. Additionally, this connection method may increase the likelihood of hallucinations produced by LLMs and increase the token cost of LLMs, thus affecting the final impact on idea evaluation and the actual expenses.

## G   SCALABILITY GENERALIZATION

To validate the generalization capability of GraphEval-GNN on large-scale datasets, we conducted experiments on the ASAP-Review dataset (Yuan et al., 2022). The ASAP-Review dataset is an open peer review dataset that includes 5,192 ICLR papers from 2017-2020 obtained through OpenReview and 3,685 NeurIPS papers from 2016-2019 accessed through NeurIPS Proceedings. A detailed introduction to this dataset, along with its composition, can be found in Section 3.1 and Table 2 of (Yuan et al., 2022). Similar to the settings described in Section 6 of our paper, we used the abstracts of all papers in the dataset as inputs and the review decisions of the papers as the predicted labels, which included Accept (Oral), Accept (Spotlight), Accept (Poster), and Reject. We divided the dataset into training, validation, and test sets in the proportions of 70%, 10%, and 20%, respectively. It is important to note that for NeurIPS papers, since only accepted papers are included and no specific labels such as Oral, Spotlight, or Poster and ratings are provided, we have to assign all paper labels as Accept (Poster). This approach ensures the accuracy of the sample because over 85% of the papers accepted at the NeurIPS conference are designated as posters. As shown in Table 24, we compared the performance of GraphEval-GNN with that of Fine-tuned BERT and Prompted LLM on this dataset. We observed that GraphEval-GNN still maintains the best performance on this large-scale dataset, with an accuracy 9.8% better than the strongest baseline, Fine-tuned BERT. Furthermore, although the rare labels of Accept (Oral) and Accept (Spotlight) (less than 4%) make it difficult for all methods to perform well in terms of macro F1 score, GraphEval-GNN still achieved an 8% improvement in macro F1 score compared to Fine-tuned BERT. These observations demonstrate the robust generalization capability of GraphEval-GNN on large-scale datasets.

## H   ACCURACY EVALUATION OF VIEWPOINTS

In order to evaluate the accuracy of viewpoints generated from ideas, we explore from two perspectives. First, we use a prompt-based approach (Luo et al., 2023; Gao et al., 2023), allowing a large LLM to assess whether each viewpoint is consistent with the original idea. Specifically, we employ the LLaMa-3.1 (405b) LLM[5], which has shown excellent performance in evaluation tasks, as the evaluator. Using the prompt from Table 25, we evaluate the consistency between the viewpoint and the idea, with an output of 1 indicating consistency and 0 indicating inconsistency. We calculate the proportion of samples judged consistent and average this across all samples to determine the consistency rate. We finally achieve consistency rates of 99.47% and 99.82% for the ICLR Papers and AI Researcher datasets, respectively. These rates, very close to 100%, demonstrate the high degree of consistency between the generated viewpoints and the original ideas as achieved by our method.

Additionally, we measure the accuracy of viewpoints from an entity-level perspective. Specifically, we first aggregate the constructed viewpoints and then assess their entity-level accuracy with respect to the idea using entity-level factual consistency metrics (Nan et al., 2021). We report the results on the datasets ICLR Papers and AI Researcher in Table 26. From the table, we can observe that the entity-level Precision, Recall, and F1 Score between the viewpoints and the idea exceed 0.9 on both datasets, which also validates the accuracy and rationality of our viewpoints.

---

[5]https://ai.meta.com/blog/meta-llama-3-1/

Table 11: **CoT prompt template**. We modify the prompt used for prompted LLM to adopt a CoT format, guiding it to complete the idea evaluation step by step.

[Instruction]
Please evaluate the paper draft step by step based on the following dimensions. For each step, carefully think through and evaluate the corresponding dimension, and then provide ratings for each dimension (1-10). You must give an overall score (0-100) along with the 6 dimension scores. No detailed analysis is needed, but ensure that your evaluation for each step is based on logical reasoning.
[Input]
Here is the paper draft to evaluate: Title – {title}; Abstract – {abstract};
[Step 1: Evaluate Novelty]
First, evaluate the novelty of the paper. {text for novelty in the idea evaluation criteria prompt template.}
Novelty Rating (1-10):
[Step 2: Evaluate Validity]
Next, evaluate the validity of the paper. {text for validity in the idea evaluation criteria prompt template.}
Validity Rating (1-10):
[Step 3: Evaluate Significance]
Then, evaluate the significance of the paper. {text for significance in the idea evaluation criteria prompt template.}
Significance Rating (1-10):
[Step 4: Evaluate Rigorousness]
Now, evaluate the rigorousness of the paper. {text for rigorousness in the idea evaluation criteria prompt template.}
Rigorousness Rating (1-10):
[Step 5: Evaluate Clarity]
Next, evaluate the clarity of the paper. {text for clarity in the idea evaluation criteria prompt template.}
Clarity Rating (1-10):
[Step 6: Evaluate Ethical Considerations]
Lastly, evaluate the ethical considerations of the paper. {text for ethnic in the idea evaluation criteria prompt template.}
Ethical Considerations Rating (1-10):
[Step 7: Final Overall Score]
After completing all the dimension evaluations, summarize your assessment and give an overall score that reflects the paper's general quality and performance across all dimensions.
Overall Score (0-100):
[Step 8: Final Decision]
Based on the overall score and individual ratings, choose the most appropriate review decision. Carefully consider how the paper performs in each dimension, and select from the following categories:
{idea evaluation decision prompt template}
Decision:
**Note:** The approximate distribution of decisions for papers at this ML venue is as follows: {label distribution of the dataset}. Please take this decision distribution into account and make your judgment carefully.
[Examples for Evaluation Standards]
{one example per decision}
[Output]
The output format should follow these rules:
Novelty Rating (1-10):
Validity Rating (1-10):
Significance Rating (1-10):
Rigorousness Rating (1-10):
Clarity Rating (1-10):
Ethical Considerations Rating (1-10):
Overall Score (0-100):
Decision: {one decision from "Reject", "Accept (Poster)", "Accept (Oral)", "Accept (Spotlight)"}

Table 12: **ToT prompt template: Novelty Evaluation Step**

[Instruction]
Please evaluate the novelty of the paper draft provided.
`{text for novelty in the idea evaluation criteria prompt template.}`
You only need to give a novelty rating (0-10). No detailed analysis is required.
[Input]
Title: `{title}`
Abstract: `{abstract}`
[Output]
Please generate a rating for the novelty of this paper (1-10)
An example of the output:
Novelty Rating (1-10): 5

Table 13: **ToT prompt template: Validity Evaluation Step**

[Instruction]
Please evaluate the validity of the paper draft based on the provided title, abstract, and novelty rating.
`{text for validity in the idea evaluation criteria prompt template.}`
You only need to give a novelty rating (0-10). No detailed analysis is required.
[Input]
Title: `{title}`
Abstract: `{abstract}`
Novelty Rating (1-10): `{novelty rating}`
[Output]
Please generate a rating for the validity of this paper (1-10)
An example of the output:
Validity Rating (1-10): 5

Table 14: **ToT prompt template: Overall Score Step**

[Instruction]
Please evaluate the overall quality of the paper draft based on the provided title, abstract, and ratings (novelty, validity, significance, rigorousness, clarity, and ethical considerations).
The overall score should reflect the general quality of the paper and how well it performs across all the evaluation dimensions.
You only need to give an overall score (0-100). No detailed analysis is required.
[Input]
Title: `{title}`
Abstract: `{abstract}`
Novelty Rating (1-10): `{novelty result}`
Validity Rating (1-10): `{validity result}`
Significance Rating (1-10): `{significance result}`
Rigorousness Rating (1-10): `{rigorousness result}`
Clarity Rating (1-10): `{clarity result}`
Ethical Considerations Rating (1-10): `{ethical considerations result}`
[Output]
Please generate an overall score for this paper (0-100).
An example of the output:
Overall Score (0-100): 80

Table 15: **ToT prompt template: Finale Decision Step**

---

[Instruction]
Please determine the final decision for the provided paper draft based on the provided title, abstract, overall score, and individual ratings (novelty, validity, significance, rigorousness, clarity, and ethical considerations). The decision should reflect the overall quality of the paper and how well it performs across all evaluation dimensions. Select the most appropriate option from the following four categories:
`{idea evaluation decision prompt template}`
**Note:** The approximate distribution of decisions for papers at this ML venue is as follows: `{label distribution of the dataset}`. Please take this decision distribution into account and make your judgment carefully.
[Examples for Evaluation Standards]
`{one example per decision}`
[Input]
Title: `{title}`
Abstract: `{abstract}`
Novelty Rating (1-10): `{novelty result}`
Validity Rating (1-10): `{validity result}`
Significance Rating (1-10): `{significance result}`
Rigorousness Rating (1-10): `{rigorousness result}`
Clarity Rating (1-10): `{clarity result}`
Ethical Considerations Rating (1-10): `{ethical considerations result}`
Overall Score (0-100): `{overall score}`
[Output]
Decision: {one decision from "Reject", "Accept (Poster)", "Accept (Oral)", "Accept (Spotlight)"}
An example of the output:
Decision: Accept (Poster)

---

Table 16: **Research Agent prompt template: Problem Validation Step**. Criteria is presented in Table 10 of the paper by Baek et al. (2024).

[System Message]
You are an AI assistant whose primary goal is to summarize the research problem in an academic paper based on its title and abstract, and to assess the quality and validity of the research problem across various dimensions. Your evaluations and feedback will help researchers refine their research problems, thereby enhancing the impact and scope of their work.

[User Message]
You will be provided with the title and abstract of an academic paper, and you need to extract its research problem and the rationale for the research problem. You are required to evaluate the research problem based on the following dimensions: Clarity, Relevance, Originality, Feasibility, and Significance, with a focus on whether it is clearly, accurately, and understandably defined.

The academic paper title and abstract to be evaluated are as follows:
Paper Title: {title}
Paper Abstract: {abstract}
Now, please proceed with a systematic evaluation focusing on Clarity, Relevance, Originality, Feasibility, and Significance:
- First, carefully read the provided title and abstract, and extract the research problem and its rationale.
- Next, generate a review and feedback that is constructive, helpful, and concise, focusing on the research problem's Clarity, Relevance, Originality, Feasibility, and Significance.
- Finally, rate the problem on a 5-point Likert scale, with 1 being the lowest score. Ensure that your ratings are discerning and critical to avoid uniformly high scores (4-5) unless fully justified. The definitions for each evaluation criterion are as follows:
{criteria}
Output:
First, summarize the research problem and its rationale from the provided paper. After evaluating the content, provide your review, feedback, and ratings in the following format:
Research Problem: {research problem}
Rationale: {research problem rationale}
Review: {review}
Feedback: {feedback}
Rating (1-5): Clarity-{rating} Relevance-{rating} Originality-{rating} Feasibility-{rating} Significance-{rating}

Table 17: **Research Agent prompt template: Method Validation Step**. `Criteria` is presented in Table 10 of the paper by Baek et al. (2024).

---

[System Message]
You are an AI assistant whose primary goal is to summarize the scientific method used in a research paper based on its title and abstract, and to evaluate the quality and soundness of the method across various dimensions. Your feedback will help researchers refine their methods, thereby enhancing the impact and reach of their work.
[User Message]
You will be provided with the title and abstract of an academic paper. From this, you are required to summarize its Scientific Method and Scientific Method Rationale. You need to evaluate the method for its Clarity, Validity, Rigorousness, Innovativeness, and Generalizability, focusing on whether the method is described clearly, precisely, and understandably, ensuring that it can be replicated and easily comprehended.
As part of your evaluation, you may refer to the research problem of the paper, which will help you better understand the context of the method and conduct a more comprehensive assessment.
The academic paper title and abstract to be evaluated and the research problem are as follows:
Paper Title: {title}
Paper Abstract: {abstract}
Research problem: {research problem}
Rationale: {research problem rationale}
Now, please proceed with the systematic evaluation of the method based on Clarity, Validity, Rigorousness, Innovativeness, and Generalizability:
- First, carefully read the provided paper title and abstract, keeping in mind the context provided by the research problem, and summarize the scientific method and its rationale.
- Next, generate a review and feedback that should be constructive, helpful, and concise, focusing on the method's Clarity, Validity, Rigorousness, Innovativeness, and Generalizability.
- Finally, provide ratings on a 5-point Likert scale, with 1 being the lowest. Ensure that your ratings are discerning and critical, avoiding a tendency toward uniformly high scores (4-5) unless fully justified. The definitions of each evaluation criterion are as follows:
{criteria}
Output:
First, summarize the scientific method and its rationale. After evaluating the content, please provide your review, feedback, and ratings in the following format:
Scientific Method: {scientific method}
Rationale: {scientific method rationale}
Review: {review}
Feedback: {feedback}
Rating (1-5): Clarity-{rating} Validity-{rating} Rigorousness-{rating} Innovativeness-{rating} Generalizability-{rating}

---

Table 18: **Research Agent prompt template: Experiment Validation Step**. `Criteria` is presented in Table 10 of the paper by Baek et al. (2024).

[System Message]
You are an AI assistant whose primary goal is to summarize the experimental design in an academic paper based on its title and abstract and meticulously evaluate the experimental design across various dimensions. Your evaluations and feedback will help researchers refine their experimental approaches, thereby amplifying the quality and impact of their scientific contributions.

[User Message]
You will be provided with the title and abstract of an academic paper. From this, you are required to summarize its experiment design and experiment design rationale. You are going to evaluate the experiment design for its Clarity, Validity, Robustness, Feasibility, and Reproducibility in validating a scientific method to address a research problem, focusing on how well it is described in a clear, precise, and understandable manner, enabling others to grasp the setup, procedure, and expected outcomes.

As part of your evaluation, you can refer to the research problem and scientific method, which will help in understanding the context of the designed experiment for a more comprehensive assessment.

The academic paper title and abstract to be evaluated, along with the research problem and scientific method, are as follows:

Paper Title: {title}
Paper Abstract: {abstract}
Research problem: {research problem}
Rationale: {research problem rationale}
Scientific Method: {scientific method}
Rationale: {scientific method rationale}

Now, proceed with your systematic evaluation of Clarity, Validity, Robustness, Feasibility, and Reproducibility:

- Start by thoroughly reading the provided paper title and abstract, keeping in mind the context provided by the research problem and scientific method mentioned above. Summarize the experiment design and its rationale.
- Next, generate a review and feedback that should be constructive, helpful, and concise, focusing on the Clarity, Validity, Robustness, Feasibility, and Reproducibility of the experiment.
- Finally, provide ratings on a 5-point Likert scale, with 1 being the lowest. Ensure that your evaluation is discerning and critical, avoiding a tendency toward uniformly high scores (4-5) unless fully justified:

{criteria}

Output:

First, summarize the experiment design and its rationale. After evaluating the content, please provide your review, feedback, and ratings in the following format:

Experiment Design: {experiment design}
Rationale: {experiment design rationale}
Review: {review}
Feedback: {feedback}
Rating (1-5): Clarity-{rating} Validity-{rating} Robustness-{rating} Feasibility-{rating} Reproducibility-{rating}

Table 19: **Research Agent prompt template: Finale Decision Step**. Building on the work of Baek et al. (2024), we further introduce a final decision step that synthesizes the evaluation results from the aforementioned steps to provide a comprehensive review decision.

---

You are an AI assistant. You will be provided with the title and abstract of an academic paper, along with a summary of its research problem, scientific method, and experiment design. Additionally, you will receive reviews, feedback, and ratings (on a scale of 1-5) for the research problem, scientific method, and experiment design across various dimensions.

Based on the provided paper title and abstract, as well as the evaluations of its research problem, scientific method, and experiment design, your task is to assign an overall score (0-100) to the paper.

You will also classify the paper into one of the following four categories based on the evaluation:

`{idea evaluation decision prompt template}`

**Note:** The approximate distribution of decisions for papers at this ML venue is as follows:
`{label distribution of the dataset}`. Please take this decision distribution into account and make your judgment carefully.

[Examples for Evaluation Standards]
`{one example per decision}`

[Input]
Paper Title: `{title}`
Paper Abstract: `{abstract}`
Research Problem: `{research problem}`
Research Problem Rationale: `{research problem rationale}`
Research Problem Review: `{research problem review}`
Research Problem Feedback: `{research problem feedback}`
Research Problem Rating: `{research problem rating}`
Scientific Method: `{scientific method}`
Scientific Method Rationale: `{scientific method rationale}`
Scientific Method Review: `{scientific method review}`
Scientific Method Feedback: `{scientific method feedback}`
Scientific Method Rating: `{scientific method rating}`
Experiment Design: `{experiment design}`
Experiment Design Rationale: `{experiment design rationale}`
Experiment Design Review: `{experiment design review}`
Experiment Design Feedback: `{experiment design feedback}`
Experiment Design Rating: `{experiment design rating}`
[Output]
You only need to give an overall score (0-100) and select a review decision. No detailed analysis is required. The output format should follow these rules:
Overall Score (0-100)= `{score}`
`{one decision from "Reject", "Accept (Poster)", "Accept (Oral)", "Accept (Spotlight)"}`
An example of the output:
Overall Score (0-100)= 82
Reject

---

Table 20: **Comparative performance results on the Fact Verification dataset.** Bold text denotes the best results. For all metrics—Accuracy, Macro Precision, Macro Recall, and Macro F1 Score—higher values indicate more precise predictions.

| Model | Accuracy | Precision | Recall | F1 Score |
|---|---|---|---|---|
| Prompted LLM (7B) | 49.79% | 57.19% | 52.27% | 47.59% |
| Prompted LLM (72B) | 59.52% | 63.13% | 60.35% | 56.33% |
| Finetuned-Bert | 70.27% | 69.74% | 68.54% | 68.64% |
| GraphEval-LP | 82.83% | 83.41% | 83.04% | 82.40% |
| **GraphEval-GNN** | **85.00%** | **90.00%** | **83.00%** | **84.00%** |

Table 21: **Comparative performance results under the setting of idea evaluation of different years.** Bold text denotes the best results. For all metrics—Accuracy, Macro Precision, Macro Recall, and Macro F1 Score—higher values indicate more precise predictions.

| Model | Accuracy | Precision | Recall | F1 Score |
|---|---|---|---|---|
| Prompted LLM (7B) | 16.67% | 20.63% | 26.12% | 18.25% |
| Prompted LLM (72B) | 14.29% | 11.25% | 32.47% | 11.76% |
| Finetuned-Bert | 48.41% | 42.46% | 36.14% | 31.57% |
| GraphEval-LP | 63.20% | 52.38% | 48.60% | 44.72% |
| **GraphEval-GNN** | **76.19%** | **48.25%** | **57.38%** | **51.32%** |

Table 22: **Performance comparison of different lightweight graph models.**

| Model | Accuracy | Precision | Recall | F1 Score |
|---|---|---|---|---|
| GraphEval-SGC | 61.0% | 27.7% | 23.3% | 27.3% |
| GraphEval-LightGCN | 54.0% | 23.43% | 25.05% | 26.70% |
| **GraphEval-GNN** | **76.0%** | **38.40%** | **37.30%** | **44.80%** |

Table 23: **Performance comparison of GraphEval-GNN via two different alternative relation extraction methods.**

| Model | Accuracy | Precision | Recall | F1 Score |
|---|---|---|---|---|
| Hybrid | 62.0% | 25.08% | 27.60% | 25.46% |
| **GraphEval-GNN** | **76.0%** | **38.40%** | **37.30%** | **44.80%** |

Table 24: **Comparative performance results for different models on the ASAP-Review dataset.** Bold text denotes the best results. For all metrics—Accuracy, Macro Precision, Macro Recall, and Macro F1 Score—higher values indicate more precise predictions.

| Model | Accuracy | Precision | Recall | F1 Score |
|---|---|---|---|---|
| Prompted LLM (7B) | 22.00% | 11.04% | 28.57% | 12.83% |
| Prompted LLM (72B) | 4.00% | 4.00% | 17.86% | 3.04% |
| Finetuned-Bert | 61.17% | 29.81% | 30.37% | 29.86% |
| **GraphEval-GNN** | **67.02%** | **33.11%** | **32.86%** | **32.20%** |

Table 25: **Prompt template of viewpoint accuracy evaluation.**

[Instruction]
Decide if the following Viewpoint, derived from the idea, is consistent with the Idea. Note that consistency means all information in the viewpoint is fully supported by the idea.
[Input]
Idea: {idea}
Viewpoint: {viewpoint}
[Output]
Explain your reasoning step by step, identifying if each part of the viewpoint aligns with the idea, then answer: Is the viewpoint consistent with the idea? Answer with only 1 for yes or 0 for no.

Table 26: **Performance of entity-level factual consistency metrics for ICLR Papers and AI Researcher datasets.**

| Dataset | Precision | Recall | F1 Score |
|---|---|---|---|
| ICLR Papers | 0.9339 | 0.9288 | 0.9314 |
| AI Researcher | 0.9472 | 0.9004 | 0.9232 |

