# OpenReview forum: "GraphEval: A Lightweight Graph-Based LLM Framework for Idea Evaluation"
_ICLR.cc/2025/Conference — ICLR 2025 Poster_

### Official Review · Reviewer_5M8a · 2024-10-29

**Soundness:** 3
**Presentation:** 3
**Contribution:** 2
**Rating:** 6
**Confidence:** 4

**Summary:**

The paper presents a novel framework that leverages graph structures to enhance large language models in the evaluation of complex research ideas. The authors propose two core methods: GraphEval-LP, a label propagation-based approach, and GraphEval-GNN, a GNN model for predicting evaluation scores. The framework addresses known limitations in LLM-based evaluation, such as sensitivity to prompts and biases toward positive feedback, by breaking down ideas into viewpoint nodes and constructing graphs to propagate scores across these nodes. Experiments on two datasets demonstrate that GraphEval improves F1 scores compared to baselines, while also detecting plagiarized ideas.

**Strengths:**

1. The introduction of graph-based methods to enhance LLMs in research idea evaluation is innovative.
2. The methodology is explained in detail, with clear steps for viewpoint extraction, relation identification, and the integration of label propagation and GNN techniques.
3. The results demonstrate significant improvements in F1 scores over multiple baselines.

**Weaknesses:**

1. While the approach works well on specific datasets (ICLR Papers and AI Researcher), the generalizability to other domains or different types of research content is unclear. Given the topic of the paper is research idea validation, the experiment turned out to detect whether this paper will be accepted. The claim and experiment are a bit stretched.
2. The paper mentions that the LLM-based relation extraction yields sparse edges between viewpoint nodes (with a 10.73% edge density on average). This sparsity may limit the effectiveness of GraphEval in certain scenarios, particularly where relationships between ideas are less explicit. The authors address this by using BERT embeddings, but further experiments with alternative relation extraction methods could improve robustness.
3. While GraphEval-GNN offers significant improvements in performance, its scalability is questionable for larger datasets. Training GNNs on extensive viewpoint-subgraphs may become computationally prohibitive, especially if the graph structure grows significantly.

**Questions:**

Is it possible to try alternative methods for relation extraction, such as unsupervised learning or hybrid approaches, which might address the sparsity issue in graph construction, especially when dealing with less structured idea content?

---

> ### Author Response · Authors · 2024-11-21
> **Response to Reviewer 5M8a (1/2)**
>
> **Q1. While the approach works well on specific datasets (ICLR Papers and AI Researcher), the generalizability to other domains or different types of research content is unclear. Given the topic of the paper is research idea validation, the experiment turned out to detect whether this paper will be accepted. The claim and experiment are a bit stretched.**
>
> **Response:** Thanks for the reviewer’s constructive feedback. We believe that research idea validation is an important and urgent topic to study. Existing tasks or datasets for idea evaluation often rely on employing human evaluators to assign labels to ideas, a method that is not only costly but also makes it difficult to acquire large training datasets. Therefore, we focus on tasks and datasets from open reviews, such as those at ICLR. Given that ICLR reviewers are generally of high caliber, this dataset not only yields high-quality labels but also provides a substantial amount of free data. The research tasks associated with these datasets are also of widespread academic interest. Additionally, we focus on whether papers are accepted for two reasons: 1. This data is relatively easy to obtain and abundantly available; 2. Other tasks, such as predicting the average score of papers, can be subjective due to varying scoring standards among reviewers, whereas the acceptance or rejection of papers tends to be more objective.
>
> On the other hand, inspired by your feedback and Reviewer zpR7's Q1, we attempt to validate our method in a long-form text evaluation task to test its generalizability to other domains or different types of research content. Specifically, we used human-annotated data from the FActScore dataset [1], where each entry contains "atomic facts" about celebrities generated by LLMs, along with assessments from human annotators on whether these "atomic facts" were supported by the materials provided to the annotators. Based on the "atomic facts" and human annotations from the training set, our method needed to predict the labels of "atomic facts" in the test set that were partitioned off. We selected topics such as Ramesses IV, Lanny Flaherty, and Florencia Bertotti, and divided the training, validation, and test sets in a 7:1:2 ratio. We compared GraphEval and some applicable baselines on this dataset in the following table. The experimental results in the table verify that our approach performs well on the long-form text evaluation task, demonstrating good adaptability to various tasks. All the discussions and experimental results mentioned above have been updated in Section E.1 of the Appendix and Table 20 of the revised PDF version.
>
> **Comparative performance results on the Fact Verification dataset.** Bold text denotes the best results.
>
> | Model              | Accuracy | Precision | Recall | F1 Score |
> |--------------------|----------|-----------|--------|----------|
> | Prompted LLM (7B)  | 49.79%   | 57.19%    | 52.27% | 47.59%   |
> | Prompted LLM (72B) | 59.52%   | 63.13%    | 60.35% | 56.33%   |
> | Finetuned-Bert     | 70.27%   | 69.74%    | 68.54% | 68.64%   |
> | GraphEval-LP       | 82.83%   | 83.41%    | 83.04% | 82.40%   |
> | **GraphEval-GNN**  | **85.00%** | **90.00%** | **83.00%** | **84.00%** |
>
> **[1]** FActScore: Fine-grained Atomic Evaluation of Factual Precision in Long Form Text Generation, EMNLP 2023.
>
> **Q2. While GraphEval-GNN offers significant improvements in performance, its scalability is questionable for larger datasets. Training GNNs on extensive viewpoint-subgraphs may become computationally prohibitive, especially if the graph structure grows significantly.**
>
> **Response:** Thanks for your valuable feedback. Indeed, in some standard and mature GNN tasks such as OGB [1,2], the number of nodes and edges can reach the order of hundreds of millions. In real-world large-scale deployment scenarios like recommendation systems, GNNs can efficiently handle tens of billions of nodes and trillions of edges [3,4]. The number of nodes in our viewpoint-subgraphs often does not reach the million level, so technically, training GNNs on extensive viewpoint-subgraphs is not problematic.
>
> Furthermore, our paper also provides another method that performs slightly worse than GraphEval-GNN but is more efficient and computationally friendly, called GraphEval-LP. This method is based on label propagation and has a linear time complexity of $O(k \cdot n \cdot p) \ll O(n^2)$, where $k$ is the number of iterations, $p$ is the predefined maximum degree for each node, and $n$ is the number of nodes. Therefore, as the number of viewpoint-subgraphs increases, GraphEval-LP can efficiently perform idea evaluation.
>
> **[1]** Open graph benchmark: Datasets for machine learning on graphs, Neurips 2020.
>
> **[2]** Graph neural networks: A review of methods and applications, AI Open 2020.
>
> **[3]** Graph convolutional neural networks for web-scale recommender systems, KDD, 2018.
>
> **[4]** Pregel: a system for large-scale graph processing, KDD 2010.

---

> > ### Author Response · Authors · 2024-11-21
> > **Response to Reviewer 5M8a (2/2)**
> >
> > **Q3. The paper mentions that the LLM-based relation extraction yields sparse edges between viewpoint nodes (with a 10.73% edge density on average). This sparsity may limit the effectiveness of GraphEval in certain scenarios, particularly where relationships between ideas are less explicit. The authors address this by using BERT embeddings, but further experiments with alternative relation extraction methods could improve robustness. Is it possible to try alternative methods for relation extraction, such as unsupervised learning or hybrid approaches, which might address the sparsity issue in graph construction, especially when dealing with less structured idea content?**
> >
> > **Response:** Thanks for the reviewer’s insightful feedback. We answer your question from two aspects.
> >
> > **[The validity and robustness of embedding similarity method]** In fact, the relation extraction method based on embedding similarity can not only construct the viewpoint-node relationship efficiently and at low cost but also can flexibly cope with various tasks by adjusting different edge densities to improve its robustness across different tasks. Specifically, as the response to Q2 of Reviewer d4yk, to explore the impact of different edge densities (as defined in Section 3, where the number of edges is divided by the proportion of all possible edge connections) on the performance of GraphEval-GNN, we selected five groups of varying edge densities for experimentation and obtained their final results, as shown in the table below. From the table, it can be observed that the performance of GraphEval-GNN initially increases and then decreases as the edge density rises. This suggests that an increase in edge density can enhance the representation of GNNs in the idea evaluation problem to a certain extent, but excessively high edge density may lead to performance degradation due to oversmoothing [1, 2]. In practical applications, we can adjust the performance at different edge densities on the validation set to enhance the robustness of our method across various tasks and datasets.
> >
> > **Performance under different edge densities**
> >
> > The performance of GraphEval-GNN initially increases and then decreases as the edge density rises.
> >
> > | Edge Density | Accuracy | Precision | Recall | F1 Score |
> > |--------------|----------|-----------|--------|----------|
> > | 5.6%         | 53.2%    | 14.54%    | 18.77% | 19.76%   |
> > | 11.4%        | 64.6%    | 20.52%    | 26.86% | 27.77%   |
> > | 22.7%        | 72.2%    | 35.98%    | 36.74% | 43.68%   |
> > | **45.5%**    | **76.0%**| **38.40%**| **37.30%**| **44.80%** |
> > | 91%          | 66.5%    | 22.76%    | 30.22% | 29.37%   |
> >
> > **[Hybrid relation extraction method]** We also followed the reviewer's suggestion and tried to propose a hybrid relation extraction method named Hybrid to compare with our fully similarity-based approach, GraphEval. Specifically, the hybrid method uses Prompted LLMs mentioned in Section 3 to connect nodes within viewpoint-subgraphs, while the edges between viewpoint-subgraphs are still based on similarity. The results of the two relation extraction methods on the ICLR Papers dataset are presented in the following table, showing that GraphEval-GNN performs better than Hybrid. This might be due to the difficulty of ensuring adequate edge density when connecting nodes within viewpoint-subgraphs using Prompted LLMs. Additionally, this connection method may increase the likelihood of hallucinations produced by LLMs and increase the token cost of LLMs, thus affecting the final impact on idea evaluation and the actual expenses. All the discussions and experimental results mentioned above have been updated in Section F.3 of the Appendix and Table 24 of the revised PDF version.
> >
> > **Performance comparison of GraphEval-GNN via two different alternative relation extraction methods.**
> >
> > | Model          | Accuracy | Precision | Recall | F1 Score |
> > |----------------|----------|-----------|--------|----------|
> > | Hybrid         | 62.0%    | 25.08%    | 27.60% | 25.46%   |
> > | **GraphEval-GNN** | **76.0%** | **38.40%** | **37.30%** | **44.80%** |
> >
> > **[1]** Measuring and relieving the over-smoothing problem for graph neural networks from the topological view, AAAI 2020
> >
> > **[2]** A survey on oversmoothing in graph neural networks, arXiv preprint 2023.

---

> > > ### Author Response · Authors · 2024-11-24
> > > **Could you let us know if our rebuttal has sufficiently addressed your concerns?**
> > >
> > > Dear Reviewer 5M8a,
> > >
> > > We recognize that the timing of this discussion period may not align perfectly with your schedule, yet we would greatly value the opportunity to continue our dialogue before the deadline approaches.
> > >
> > > We hope that our responses and additional experiments have effectively addressed your concerns. We truly appreciate all the valuable advice we have received, and we are pleased to share that one of the reviewers has kindly recognized our improvements by raising their score. This acknowledgment reflects the positive impact of our collaborative efforts in enhancing the quality of the paper.
> > >
> > > Could you let us know if your concerns have been adequately addressed? If you find that your concerns have been resolved, we would appreciate it if you could reconsider the review score.
> > >
> > > Thanks!

---

> > > > ### Author Response · Authors · 2024-11-25
> > > > **Looking Forward to Further Discussion**
> > > >
> > > > Dear reviewer 5M8a,
> > > >
> > > > As the discussion period ends soon, we would like to check whether our responses answer your questions. Following your insightful comments, we have clarified the rationale and academic value of focusing on tasks in datasets such as ICLR Papers, as well as the scalability capabilities of GraphEval-GNN. We have also conducted experiments on the long-form text evaluation task to test the generalizability of GraphEval to other domains or different types of research content and proposed a hybrid relation extraction method, which we have discussed and compared with GraphEval. Thank you again for your comments and suggestions to improve our paper, and we look forward to your reply.
> > > >
> > > > Best,
> > > > Authors

---

> > > > > ### Comment · Reviewer_5M8a · 2024-11-26
> > > > >
> > > > > Thanks for your response. After reading the rebuttal, I believe the authors still do not fully address my concerns. The "research idea validation" task is more like a reasoning task for me. It evaluates whether the main idea in a paper is valid or not. This is not equivalent to "paper acceptance." Whether a paper is accepted can be based on multiple factors, like the overall presentation, experiments, etc.
> > > > >
> > > > > I'll increase the score by 1.

---

> > > > > > ### Author Response · Authors · 2024-11-27
> > > > > > **Clarification for our experiment settings and express sincere thanks for the reviewer's constructive feedback**
> > > > > >
> > > > > > Thank you for the reviewer's insightful feedback and hard work. Your valuable suggestions have significantly improved our paper. However, we still want to clarify that our primary focus is on idea evaluation, and the abstracts of papers reflect the core ideas, main methods, and experiments of the entire paper, serving as a comprehensive summary that includes multiple factors. Therefore, we have chosen abstracts as the representative element for idea evaluation. We also acknowledge that other specific factors mentioned by the reviewer may influence paper acceptance. However, our paper primarily focuses on the general study of idea evaluation rather than the specific area of predicting paper acceptance. We will discuss these aspects in detail in the future work part of the revised PDF.

---

### Official Review · Reviewer_d4yk · 2024-11-01

**Soundness:** 3
**Presentation:** 3
**Contribution:** 4
**Rating:** 8
**Confidence:** 4

**Summary:**

This paper proposes GraphEval, a lightweight graph-based LLM framework for idea evaluation, which utilizes LLM's summarization and abstraction capabilities for improving idea evaluation. Through the viewpoint graph extraction, label propagation or GNN algorithms, and plagiarism detection mechanism, GraphEval breaks down complex ideas into multiple simple viewpoints, connects the viewpoints into an entire viewpoint-graph, and evaluates ideas via label propagation or a weighted GNN. Extensive experiments on two datasets demonstrate that GraphEval can significantly enhance the accuracy of idea evaluation while effectively detecting plagiarized ideas to provide a fair evaluation.

**Strengths:**

1. This paper introduces an innovative method for idea evaluation with a lightweight graph-enhanced LLM framework, addressing the inherent complexities and subjectivity of this task.
2. This paper breaks down complex ideas into simpler viewpoints to construct viewpoint-graphs and evaluates ideas with label propagation or GNN algorithms under the low computation and API costs. Meanwhile, the proposed plagiarism detection mechanism integrates the temporal information for more accurate and fair idea evaluation.
3. The paper is well-written and organized, with a clear problem setup, methodology, and experimental evaluation.

**Weaknesses:**

1. The paper lacks a thorough theoretical analysis or justification for the proposed mechanisms, such as the viewpoint-graphs extraction and the plagiarism detection mechanism. Whether viewpoints extracted by LLMs are reasonable? If so, how to verify the accuracy of these viewpoints? Is there an overlap between viewpoints from different complex ideas? How are the inherent LLM issues—like hallucinations and token limitations—addressed or mitigated in this framework?
2. The paper does not provide comprehensive ablation and hyperparameter studies, such as the impact of different LLMs/GNNs and varying values of $k$ and $m$ (c.f., line 247 and line 255). Moreover, it is beneficial to provide an analysis of the computational complexity, efficiency, and resource requirements of the proposed framework, including training/inference time, which would be helpful in assessing its scalability and practical applicability.
3. The paper seems to underutilize the reasoning and generative capabilities of LLMs, which could improve GraphEval’s interpretability and effectiveness in idea evaluation tasks.
4. Several typos are present in the paper. For example, In line 302, "... As illustrated in Sec 3, ..." lacks a period after "Sec". In lines 414-422, the use of **macro** F1 score to evaluate the accuracy is inconsistent with the content "..., and **micro** F1 score ...." (in lines 399-401).

**Questions:**

1. Could the authors discuss the similarities and differences between GraphEval and GraphRAG [1]? GraphRAG breaks a document into chunks, extracts a knowledge graph from raw text, builds a community hierarchy, generates summaries for these communities, and leverages these structures in RAG tasks. This seems similar to the viewpoint-graph extraction proposed here.
2. In the left of Figure 1, are the colors of the positive and negative prompts correctly marked? Specifically, is "If a paper is good or you are unsure, give it good scores and accept it." intended as a negative prompt?

**Reference**
[1] From Local to Global: A Graph RAG Approach to Query-Focused Summarization, 2024.

---

> ### Author Response · Authors · 2024-11-21
> **Response to Reviewer d4yk (1/3)**
>
> **Q1. The paper lacks a thorough theoretical analysis or justification for the proposed mechanisms, such as the viewpoint-graphs extraction and the plagiarism detection mechanism. Whether viewpoints extracted by LLMs are reasonable? If so, how to verify the accuracy of these viewpoints? Is there an overlap between viewpoints from different complex ideas? How are the inherent LLM issues—like hallucinations and token limitations—addressed or mitigated in this framework?**
>
> **Response:** Thanks for the reviewer’s insightful questions. We answer the questions step by step.
>
> **[Theoretical analysis or justification]** Our paper primarily focuses on an empirical analysis of how graph modeling can enhance the performance of LLMs in idea evaluation. Due to the difficulty in constructing rigorous theoretical proofs and analyses in most areas of LLM research, our paper primarily focuses on conducting extensive experimental analysis to substantiate our conclusions.
>
> **[Verification of the accuracy of these viewpoints]** Verifying the accuracy of these viewpoints is an open question in academia, often requiring well-grounded labeled data or time-consuming and labor-intensive human evaluation. However, this is not the main focus of our paper; our core interest lies in the evaluation of idea quality.
>
> **[Overlap between viewpoints]** It is possible for viewpoints from different complex ideas to overlap. In our paper, this aspect is modeled using similarity edges between viewpoint nodes, where edges between overlapping viewpoint-nodes have similarity weights closer to 1.
>
> **[Solution for inherent LLM issues]** In the introduction, specifically in Figures 1 and 2, we compare the performance of LLMs and GraphEval. While current LLMs may introduce hallucinations and biases when processing complex and subjective ideas, GraphEval deconstructs these ideas into simpler, more comprehensible viewpoints. Using a graph-based framework, it provides more objective reasoning, effectively mitigating illusion issues in idea evaluation. As for token limitations, we believe this is mainly the focus of papers on long-context LLMs, not the central concern of our paper.
>
> **Q2. The paper seems to underutilize the reasoning and generative capabilities of LLMs, which could improve GraphEval’s interpretability and effectiveness in idea evaluation tasks.**
>
> **Response:** Thanks for the reviewer’s constructive feedback. Indeed, we have explored the reasoning and generative capabilities of LLMs like COT/TOT in our baselines. However, the results indicate that these capabilities currently have little impact on LLMs for idea evaluation. We believe that in the future, LLMs with larger parameters will possess more powerful reasoning and generative capabilities to effectively address the idea evaluation problem. However, these larger LLMs, compared to smaller ones, will incur higher inference costs in idea evaluation. Therefore, our framework essentially aims to utilize lightweight graphs to assist smaller LLMs in viewpoint-level reasoning, thereby enabling smaller models to achieve good results in idea evaluation.
>
>
> **Q3. Several typos are present in the paper. For example, In line 302, "... As illustrated in Sec 3, ..." lacks a period after "Sec". In lines 414-422, the use of macro F1 score to evaluate the accuracy is inconsistent with the content "..., and micro F1 score ...." (in lines 399-401).**
>
> **Response:** We apologize for the confusion. We have revised them in the current version: "micro" is corrected to "macro" and "Sec" is corrected to "Sec.".
>
> **Q4. Could the authors discuss the similarities and differences between GraphEval and GraphRAG [1]? GraphRAG breaks a document into chunks, extracts a knowledge graph from raw text, builds a community hierarchy, generates summaries for these communities, and leverages these structures in RAG tasks. This seems similar to the viewpoint-graph extraction proposed here.**
>
> **Response:** Thank you for your constructive questions. Indeed,
> the two methods seem similar, both aiming to better manage text in the form of graphs. However, they clearly differ in many aspects:
>
> **[Element]:** The constituent elements of GraphEval are viewpoints, whereas the constituent elements of GraphRAG are raw text chunks.
>
> **[Construction of the graph]:** GraphEval uses text similarity and graph algorithms, while GraphRAG employs LLMs to identify relationships, which is costly and slow.
>
> **[Applications]:** GraphEval is used to evaluate ideas and some hard-to-understand texts, while GraphRAG is designed for long-context QA.
>
> **[Technique]:** GraphEval utilizes a graph algorithm for output, whereas GraphRAG primarily uses a rag approach.

---

> > ### Author Response · Authors · 2024-11-21
> > **Response to Reviewer d4yk (2/3)**
> >
> > **Q5. The paper does not provide comprehensive ablation and hyperparameter studies, such as the impact of different LLMs/GNNs and varying values of and  (c.f., line 247 and line 255). Moreover, it is beneficial to provide an analysis of the computational complexity, efficiency, and resource requirements of the proposed framework, including training/inference time, which would be helpful in assessing its scalability and practical applicability.**
> >
> > **Response:** Thank you for the constructive feedback from the reviewers. Regarding the ablation and hyperparameter studies of the paper, we have actually discussed the selection of LLMs and the impact of different relation extraction methods in the paper. Specifically, **in Section 7.1, our experiments explored on the influence of LLMs of different sizes on the performance of idea evaluation.** We discovered that in many instances, using smaller LLMs not only cuts costs but also delivers comparable or even superior performance compared to larger models. Based on these discussions, we adopted the setting of smaller LLMs in the viewpoint extraction part of GraphEval. Additionally, **we discussed the impact and selection of relation extraction methods from the perspective of edge density in Section 3**. Indeed, because our method performs well and is robust, we did not extensively discuss specific parameter tuning and model design during the experiments. However, following the reviewers' suggestions, we conducted ablation studies on edge densities and graph neural network architectures.
> >
> > **[Impact of varying edge densities]** To explore the impact of different edge densities (as defined in Section 3, where the number of edges is divided by the proportion of all possible edge connections) on the performance of GraphEval-GNN, we selected five groups of varying edge densities for experimentation and obtained their final results, as shown in the table below. From the table, it can be observed that the performance of GraphEval-GNN initially increases and then decreases as the edge density rises. This suggests that an increase in edge density can enhance the representation of GNNs in the idea evaluation problem to a certain extent, but excessively high edge density may lead to performance degradation due to oversmoothing [1, 2]. In practical applications, we can adjust the performance at different edge densities on the validation set to enhance the robustness of our method across various tasks and datasets. All the discussions and experimental results mentioned above have been updated in Section F.3 of the Appendix and Table 24 of the revised PDF version.
> >
> > **Performance under different edge densities**
> >
> > The performance of GraphEval-GNN initially increases and then decreases as the edge density rises.
> >
> > | Edge Density | Accuracy | Precision | Recall | F1 Score |
> > |--------------|----------|-----------|--------|----------|
> > | 5.6%         | 53.2%    | 14.54%    | 18.77% | 19.76%   |
> > | 11.4%        | 64.6%    | 20.52%    | 26.86% | 27.77%   |
> > | 22.7%        | 72.2%    | 35.98%    | 36.74% | 43.68%   |
> > | **45.5%**    | **76.0%**| **38.40%**| **37.30%**| **44.80%** |
> > | 91%          | 66.5%    | 22.76%    | 30.22% | 29.37%   |
> >
> > **[Effects of various lightweight graph neural network architectures]** To compare the impact of different lightweight GNN architectures on the performance of GraphEval, we selected two classic lightweight GNN frameworks, SGC [3] and LightGCN [4], to replace the current heterogeneous graph structure in GraphEval. We named these two baselines GraphEval-SGC and GraphEval-LightGCN, respectively. We compared these baselines with GraphEval-GNN on the ICLR Papers dataset, as shown in the table below. We observed that the performance of the lightweight frameworks was inferior to that of GraphEval-GNN, which is due to their sacrifice of specific node information and simplification of the process of graph convolution modeling in order to optimize memory usage and speed. All the discussions and experimental results mentioned above have been updated in Section F.2 of the Appendix and Table 23 of the revised PDF version. In addition to the aforementioned ablation, we also evaluate the comparative impact of alternative relation extraction methods in Appendix F.3 and Table 24.
> >
> > **Performance comparison of different lightweight graph models.**
> >
> > | Model              | Accuracy | Precision | Recall | F1 Score |
> > |--------------------|----------|-----------|--------|----------|
> > | GraphEval-SGC      | 61.0%    | 27.7%     | 23.3%  | 27.3%    |
> > | GraphEval-LightGCN | 54.0%    | 23.43%    | 25.05% | 26.70%   |
> > | **GraphEval-GNN**  | **76.0%**| **38.40%**| **37.30%** | **44.80%** |

---

> > > ### Author Response · Authors · 2024-11-21
> > > **Response to Reviewer d4yk (3/3)**
> > >
> > > **Continuation of the response to Question 5**:
> > >
> > > **[Computational complexity and efficiency]** Actually, compared to many influential papers in the field of large language models (LLMs) [5,6], which do not report data on method cost and efficiency, we have already documented the token cost and price (Normed Cost) for each method in Tables 2 and 3. However, following the reviewers' suggestions, we have included a discussion on computational complexity, efficiency, and resource requirements in the revised PDF ***[lines 425-431]***. Specifically, in terms of computational complexity, we calculated the average GPU memory usage for GraphEval-GNN and Fine-tuned BERT for the two tasks, which are 372MB and 4.84 GB respectively. The detailed modifications are as follows: During the training phase, we configured the graph neural network as a two-layer weighted GNN with a hidden dimension of 64. The batch size is set to 64, and the maximum number of training epochs is limited to 1000. We employ the Adam optimizer (Diederik, 2014) for training and gradually reduce the learning rate from 1e-3 to 0 using a LambdaLR scheduler. Our proposed method is implemented using PyTorch and PyTorch Geometric, with all experiments conducted on a single NVIDIA A100 Tensor Core GPU. For the LLMs, we utilize API calls from Together AI4 to obtain responses. Additionally, the average GPU memory usage of GraphEval-GNN for the two tasks is 372MB, whereas Fine-tuned BERT utilizes 4.84 GB on average.
> > >
> > > **[1]** Chen, D., Lin, Y., Li, W., Li, P., Zhou, J., & Sun, X. (2020, April). Measuring and relieving the over-smoothing problem for graph neural networks from the topological view. In Proceedings of the AAAI conference on artificial intelligence (Vol. 34, No. 04, pp. 3438-3445).
> > >
> > > **[2]** Rusch, T. K., Bronstein, M. M., & Mishra, S. (2023). A survey on oversmoothing in graph neural networks. arXiv preprint arXiv:2303.10993.
> > >
> > > **[3]** Wu, F., Souza, A., Zhang, T., Fifty, C., Yu, T., & Weinberger, K. (2019, May). Simplifying graph convolutional networks. In International conference on machine learning (pp. 6861-6871). PMLR.
> > >
> > > **[4]** He, X., Deng, K., Wang, X., Li, Y., Zhang, Y., & Wang, M. (2020, July). Lightgcn: Simplifying and powering graph convolution network for recommendation. In Proceedings of the 43rd International ACM SIGIR conference on research and development in Information Retrieval (pp. 639-648).
> > >
> > > **[5]** Park, J. S., O'Brien, J., Cai, C. J., Morris, M. R., Liang, P., & Bernstein, M. S. (2023, October). Generative agents: Interactive simulacra of human behavior. In Proceedings of the 36th annual acm symposium on user interface software and technology (pp. 1-22).
> > >
> > > **[6]** Yao, S., Yu, D., Zhao, J., Shafran, I., Griffiths, T., Cao, Y., & Narasimhan, K. (2024). Tree of thoughts: Deliberate problem solving with large language models. Advances in Neural Information Processing Systems, 36.
> > >
> > > **Q6. In the left of Figure 1, are the colors of the positive and negative prompts correctly marked? Specifically, is "If a paper is good or you are unsure, give it good scores and accept it." intended as a negative prompt?**
> > >
> > > **Response:** We are sorry for the confusion. Our intention was to demonstrate that different prompts significantly influence LLM-based methods, thereby proving that LLM-based approaches possess strong biases in idea evaluation. Unfortunately, we inadvertently reversed the colors for positive and negative prompts. We have corrected this in the current version. Thank you again for your feedback.

---

> ### Comment · Reviewer_d4yk · 2024-11-22
> **Response to authors' rebuttal**
>
> The authors’ rebuttal addressed most of my concerns, so I have raised my score from 5 to 6.

---

> > ### Author Response · Authors · 2024-11-24
> > **Looking Forward to Further Discussion**
> >
> > Dear reviewer d4yk,
> >
> > Thank you for your prompt response. Could you kindly specify the remaining concerns? We will try our best to solve them in the next few days. And we kindly request you consider further increasing the score
> >
> > Thank you.

---

> > > ### Comment · Reviewer_d4yk · 2024-11-27
> > > **Response to authors' rebuttal**
> > >
> > > I still believe that a theoretical analysis or justification of the proposed mechanisms, such as the viewpoint-graphs extraction and the plagiarism detection mechanism, would further strengthen the paper's contribution and make it more solid.
> > >
> > > Therefore, I keep my recommendation rating of 6 for this paper.

---

> > > > ### Author Response · Authors · 2024-11-28
> > > > **Further response to Reviewer d4yk’s concern on evaluation of viewpoint-graphs extraction**
> > > >
> > > > Thanks for the reviewer’s constructive feedback. We want to reclaim that we acknowledge that the evaluation of viewpoint-graphs extraction is very important, but the main focus of our paper is on how graph modeling can enhance the performance of Large Language Models (LLMs) in idea evaluation through a series of empirical analyses. However, we follow the reviewer's suggestion and have conducted experiments to evaluate the accuracy of viewpoint-graphs extraction. Specifically, we explore from two perspectives. First, we use a prompt-based approach [1,2], allowing a large LLM to assess whether each viewpoint is consistent with the original idea. Specifically, we employ the [LLaMa-3.1 (405b) LLM](https://ai.meta.com/blog/meta-llama-3-1/), which has shown excellent performance in evaluation tasks, as the evaluator. Using the prompt from Table 1 below, we evaluate the consistency between the viewpoint and the idea, with an output of 1 indicating consistency and 0 indicating inconsistency. We calculate the proportion of samples judged consistent and average this across all samples to determine the consistency rate. We finally achieve consistency rates of 99.47% and 99.82% for the ICLR Papers and AI Researcher datasets, respectively. These rates, very close to 100%, demonstrate the high degree of consistency between the generated viewpoints and the original ideas as achieved by our method.
> > > >
> > > > Additionally, we measure the accuracy of viewpoints from an entity-level perspective. Specifically, we first aggregate the constructed viewpoints and then assess their entity-level accuracy with respect to the idea using entity-level factual consistency metrics [3]. We report the results on the datasets ICLR Papers and AI Researcher in Table 2 below. From the table, we can observe that the entity-level Precision, Recall, and F1 Score between the viewpoints and the idea exceed 0.9 on both datasets, which also validates the accuracy and rationality of our viewpoints. All the discussions and experimental results mentioned above have been updated in Section H of the Appendix and Table 26-27 of the revised PDF version.
> > > >
> > > > Could you let us know if your concerns have been adequately addressed? If you find that your concerns have been resolved, we would appreciate it if you could reconsider the review score.
> > > >
> > > > Thanks!
> > > >
> > > > **Table 1: Prompt Template of Viewpoint Accuracy Evaluation.**
> > > > ```
> > > > [Instruction]:
> > > > Decide if the following Viewpoint, derived from the idea, is consistent with the Idea. Note that consistency means all information in the viewpoint is fully supported by the idea.
> > > >
> > > > [Input]:
> > > > Idea: {idea}
> > > > Viewpoint: {viewpoint}
> > > >
> > > > [Output]:
> > > > Explain your reasoning step by step, identifying if each part of the viewpoint aligns with the idea, then answer: Is the viewpoint consistent with the idea? Answer with only 1 for yes or 0 for no.
> > > > ```
> > > >
> > > > **Table 2:Performance of entity-level factual consistency metrics for ICLR Papers and AI Researcher datasets.**
> > > >
> > > > | Dataset        | Precision | Recall | F1 Score |
> > > > | -------------- | --------- | ------ | -------- |
> > > > | ICLR Papers    | 0.9339    | 0.9288 | 0.9314   |
> > > > | AI Researcher  | 0.9472    | 0.9004 | 0.9232   |
> > > >
> > > > **[1]** ChatGPT as a Factual Inconsistency Evaluator for Text Summarization, arXiv 2023.
> > > >
> > > > **[2]** Human-like Summarization Evaluation with ChatGPT, arXiv 2023.
> > > >
> > > > **[3]** Entity-level Factual Consistency of Abstractive Text Summarization, ACL 2021.

---

> ### Comment · Reviewer_d4yk · 2024-11-28
> **Response to authors' rebuttal**
>
> Thanks for the authors' comprehensive evaluation of viewpoint-graphs extraction from two perspectives, which effectively addressed my concerns. Additionally, could authors release your related code, which ensures the reproducibility of this paper?
>
> I am happy to see the paper accepted (raising my score from 6 to 8).

---

> ### Author Response · Authors · 2024-11-28
> **Express sincere thanks for the reviewer's constructive feedback**
>
> Thank you for the reviewer's insightful feedback and hard work. Your valuable suggestions have significantly improved our paper. We are pleased to hear that our responses have addressed most of your concerns. During the rebuttal phase, we followed the reviewers' suggestions and conducted numerous new settings and experimental trials. Consequently, we are continuously organizing the code and adding necessary comments to ensure its readability and usability. We commit to including the code link in the potential camera-ready version of the paper to guarantee its reproducibility. We are committed to incorporating the suggested changes in our revisions to further enhance the manuscript. By the way, as Thanksgiving is approaching, we also sincerely wish you a Happy Thanksgiving. Thank you for the valuable feedback and hard work you provided during the rebuttal phase.

---

### Official Review · Reviewer_oRRW · 2024-11-02

**Soundness:** 2
**Presentation:** 2
**Contribution:** 2
**Rating:** 5
**Confidence:** 5

**Summary:**

This paper presents a framework addressing the limitations of LLMs in evaluating research ideas, focusing on stability, semantic comprehension, and objectivity. The authors introduce two core methods—GraphEval-LP (label propagation) and GraphEval-GNN (a GNN-based approach). Both are lightweight, with GraphEval-GNN incorporating a novelty detection component to assess plagiarism risks.

**Strengths:**

The viewpoint-graph breaks down complex ideas into interconnected, evaluable components.

**Weaknesses:**

1. Some components of model lack clear explanations.
2. The experimental evaluation is limited; for instance, there is no ablation study, and the dataset size is small.
3. The construction of the graph relies solely on textual information. Evaluating a paper’s acceptance potential based exclusively on textual relevance is not entirely reasonable.

**Questions:**

1. In the domain of automated paper review generation, several existing works relevant to this field are missing from the discussion [1-3].

2. The paper contains informal language in some variable names, such as "max_iters" and "cos sim".

3. The framework represents edges between viewpoints within a single document and across multiple documents as undirected. Could the authors clarify how (or if) temporal edges are integrated into the graph?

4. In line 288, what is $n$ when predicting the label $\hat{y}$?

5. Lines 270-274 describe the initialization process is ambiguous. The initialization node features are formed with node labels? Could the authors clarify this initialization procedure?

6. The task tackled in this study does not appear complex but the dataset used for evaluation is quite small. Given the availability of larger, relevant datasets (e.g., provided by ReviewAdvisor [1], which covers ICLR and NeurIPS), could the authors explain why a larger dataset was not employed?

7. What is the ratio of positive to negative samples in the dataset? The authors note that LLMs are inclined to accept most papers, yet the baseline methods report very low accuracy (below 20% on the ICLR dataset), suggesting a high prevalence of negative samples. Could the authors provide details on dataset composition and how they ensured a fair experimental comparison?

8. The paper lacks implementation details, and no code or datasets are available.

9. Could the authors specify the exact prompts used in the CoT and ToT baselines?

10. GraphEval framework relies on a 7B parameter LLM while Fine-tuning BERT (with much fewer parameters) achieves comparable performance. How about the performance of fine-tuning 7B models?
---

[1] Yuan, Weizhe, Pengfei Liu, and Graham Neubig. "Can we automate scientific reviewing?." Journal of Artificial Intelligence Research 75 (2022): 171-212.

[2] Du, Jiangshu, et al. "Llms assist nlp researchers: Critique paper (meta-) reviewing." arXiv preprint arXiv:2406.16253 (2024).

[3] Lin, Jialiang, et al. "Automated scholarly paper review: concepts, technologies, and challenges." Information fusion 98 (2023): 101830.

---

> ### Author Response · Authors · 2024-11-21
> **Response to Reviewer oRRW (1/3)**
>
> **Q1. In the domain of automated paper review generation, several existing works relevant to this field are missing from the discussion [1-3].**
>
> **Response:** Thank you for your feedback. In the introduction and related work sections of our paper, we have cited and discussed a vast array of representative works on idea/text evaluation. As a critically important field, idea evaluation is bound to attract widespread attention in academia, and inevitably there will be some works that we cannot discuss and cover. **Our paper currently focuses primarily on several LLM-based works, which represent the latest research trends in academia.**
>
> However, following the reviewer's suggestion, **we have added a discussion of these three automated paper review generation works in the related work section**: "In addition, numerous studies employed fine-tuned lightweight language models (e.g., BERT (Devlin, 2018)) to evaluate complex texts, such as dialogues (Thida, 2021), tweets (Pota et al., 2021), and the novelty of ideas (Just et al., 2024; Yuan et al., 2022). Conversely, recent studies have sought to leverage the domain knowledge and logical capabilities of LLMs to create idea evaluators(Ubonsiri, 2024; Baek et al., 2024; Du et al., 2024; Lin et al., 2023a). Du et al. (2024) proposed using a prompt-based approach to allow LLMs to act as reviewers and meta-reviewers in order to assess the level of papers/ideas based on different evaluation criteria.".
>
> **Q2. The paper contains informal language in some variable names, such as "max_iters" and "cos sim".**
>
> Response: Thanks for your comments. In fact, these concepts are quite basic and are often used in graph-related applications. We use this language because it facilitates the understanding of the readers. However, following the reviewer's suggestion, we have made them more formal in the revised version: "Then, we compute the cosine similarity $s$ between their embeddings: $\{s}(e_i, e_j) = \frac{e_i \cdot e_j}{\|e_i\| \|e_j\|}\$. We perform multiple iterations of label propagation on graph $G$ until the labels no longer change."
>
> **Q3. The framework represents edges between viewpoints within a single document and across multiple documents as undirected. Could the authors clarify how (or if) temporal edges are integrated into the graph?**
>
> **Response:** Thanks for your insightful question. As discussed in ***[lines 212-232] of Section 3*** of the paper, due to the sparsity of the edges constructed by the prompt-based LLM, our edges are ultimately built based on the similarity between embeddings of the viewpoint-nodes, which are undirected. We are fully aware that idea evaluation is temporal, hence in ***[lines 321-344] of Section 5***, we introduced **how we incorporate temporal information into viewpoint-nodes.** Compared to integrating temporal edges into the graph, adding temporal information to the nodes’ embeddings is simpler and more practical.
>
> **Q4. In line 288, what is n when predicting the label y^hat?**
>
> **Response:**  Thanks for your question. In ***[lines 200-203]***, we introduced that $n$ **represents the number of viewpoints extracted from a given research idea.** However, the distance from where y^hat​ is introduced is indeed too far. We are sorry for the confusion caused.
>
> In the revised version, we have added further explanation: The predicted label $\hat{y}$ is then determined by selecting the dimension with the highest value in the summed vector, i.e., $ \hat{y} = \arg\max_j \left( \sum_{i=1}^{k} d_i \right)_j $, where $j$ indexes the dimensions of the vector and $k$ means the number of viewpoints extracted from a given research idea.
>
> **Q5. Lines 270-274 describe the initialization process is ambiguous. The initialization node features are formed with node labels? Could the authors clarify this initialization procedure?**
>
> **Response:** Thanks for the reviewer’s question. We believe the reviewer has a misunderstanding about the use of label propagation in GraphEval-LP, which is a commonly used method that is simple to deploy [[1](https://en.wikipedia.org/wiki/Label_propagation_algorithm#:~:text=Label%20propagation%20is%20a%20semi,have%20labels%20(or%20classifications)].
>
> In fact, in the initialization procedure section, we do not mention node features, and the label propagation process does not involve node features either. The process of label propagation involves transferring labels from labeled viewpoint-nodes (label initialization of the training set) through the graph to unlabeled viewpoint-nodes (testing set) ***[lines 270-278]***, and aggregating the viewpoint-nodes contained in an idea ***[lines 279-283]*** to obtain the final prediction result for the idea.

---

> > ### Author Response · Authors · 2024-11-21
> > **Response to Reviewer oRRW (2/3)**
> >
> > **Q6. The task tackled in this study does not appear complex but the dataset used for evaluation is quite small. Given the availability of larger, relevant datasets (e.g., provided by ReviewAdvisor [1], which covers ICLR and NeurIPS), could the authors explain why a larger dataset was not employed?**
> >
> > **Response:**  Thank you for your question. Indeed, our paper primarily aims to demonstrate the technical contribution of GraphEval. **We have proven that GraphEval can still perform well under limited resources and data.** Following existing research on graphs (Yang et al., 2024; Zhu et al., 2024;Shang et al., 2024), we can anticipate that GraphEval will perform better with larger datasets. Moreover, we focus on the LLM field, using graphs to assist LLMs in idea evaluation. **Currently, most LLM-based idea evaluation methods are based on prompts. These methods do not improve in performance as the dataset size increases.** Therefore, to ensure a fair comparison, we have chosen  lightweight but representative datasets for our experiments.
> >
> > **Q7. What is the ratio of positive to negative samples in the dataset? The authors note that LLMs are inclined to accept most papers, yet the baseline methods report very low accuracy (below 20% on the ICLR dataset), suggesting a high prevalence of negative samples. Could the authors provide details on dataset composition and how they ensured a fair experimental comparison?**
> >
> > **Response:** We appreciate the reviewer's insightful comments. In our paper, we have shown the ratio of positive to negative samples of two datasets and the details on dataset composition in ***[Table 5 of Appendix]***. To ensure a fair experimental comparison, we made certain that all methods used consistent data ***[Table 5 of Appendix]*** and hyperparameters ***[Table 7 of Appendix]***. Furthermore, according to the task description in ***[lines 348-352]***, idea evaluation is a multi-class prediction problem. Moreover, the label distribution of the ICLR Papers Dataset closely aligns with the acceptance rate of the ICLR conference, where only about 30% of submissions are accepted. Therefore, although LLMs are inclined to accept most papers, the prompt-based LLM approach performs poorly due to significant bias in its evaluation.
> >
> > **Q8. The paper lacks implementation details, and no code or datasets are available.**
> >
> > **Response:** Thanks for your feedback. We answer your question step by step.
> >
> > **[Implementation details]** We believe that we have provided a fairly comprehensive introduction to the implementation details in various sections of the paper. The implementation details of the method are extensively introduced in the paper through detailed formulas, such as Equations (3), (4), and (5), along with corresponding descriptions and the pseudocode process of Algorithm 1 ***[lines 324-336]***. Additionally, we have listed the hyperparameter settings for GraphEval implementation in the appendix ***[Table 7]***, as well as the prompts used in GraphEval ***[Table 4 and Table 6]***. Moreover, we have thoroughly introduced the implementation of the baseline methods ***[lines 365-412]*** and enumerated all the detailed prompts used by the baseline methods ***[Table 8-19]***. Regarding the datasets and tasks, we have detailed their information in the paper ***[lines 354-362]*** and also illustrated the process of viewpoint extraction from a research idea and the specific data proportion structure in ***[Appendix A and Table 5]***.
> >
> > Moreover, in order to further improve our implementation details, we have followed the Reviewer d4yk's suggestions in Q2 and added a discussion of implementation details in ***[lines 425-431]*** of the revised version, including computational complexity, efficiency, and resource requirements.
> >
> > **[Availability of code and datasets]** Given the comprehensive details about the method implementation and datasets presented in our paper, we can ensure the reproducibility of our work. The current code and data are being further organized to enhance their readability and usability. Since ICLR does not mandatorily require the submission of data and code, we look forward to releasing the code and data in the near future.
> >
> > **Q9. Could the authors specify the exact prompts used in the CoT and ToT baselines?**
> >
> > **Response:** Thanks for your question. Actually, we have presented the exact prompts used in the CoT and ToT baselines in ***[Table 11 and 12]*** of Appendix respectively.

---

> > > ### Author Response · Authors · 2024-11-21
> > > **Response to Reviewer oRRW (3/3)**
> > >
> > > **Q10. GraphEval framework relies on a 7B parameter LLM while Fine-tuning BERT (with much fewer parameters) achieves comparable performance. How about the performance of fine-tuning 7B models?**
> > >
> > > **Response:** Thanks for your comments. We would first like to clarify that **Fine-tuned BERT does not achieve comparable performance to GraphEval**, as shown in ***[Tables 2 and 3]***. Compared to Fine-tuned BERT, GraphEval-GNN achieves **an absolute improvement of at least 13.8% and a relative improvement of at least 25.88% in F1 Score.** These data demonstrate that the performance enhancement of GraphEval over Fine-tuned BERT is very significant.
> > >
> > > Additionally, although the 7B parameter LLM has more parameters than Fine-tuned BERT, we only utilize the inference capabilities of the 7B parameter LLM. In terms of computation cost, our lightweight GraphEval has a much greater advantage. **This is because the parameter of BERT is million-level, whereas GraphEval-LP is parameter-free and GraphEval-GNN has parameters in the thousand-level.**
> > >
> > >  Lastly, **fine-tuning large parameter models, such as the 7B LLM, may indeed trade higher computation costs for performance advantages.** However, we believe that the LLM as a foundation model has already received widespread attention and research from the academic community for its zero-shot capabilities in idea evaluation (Si et al., 2024; Baek et al., 2024). Therefore, we argue that **rather than focusing on using large parameter models to fine-tune idea evaluation problems, it is more worthwhile for researchers to explore how a lightweight framework can further enhance the LLMs’ capabilities for idea evaluation. This is also the focus of our paper, **using a lightweight, graph-based framework to enhance the LLM's idea evaluation capabilities.**
> > >
> > > **Q11. The experimental evaluation is limited; for instance, there is no ablation study**
> > >
> > > **Response:** Thanks for your feedback. Regarding the ablation studies of the paper, we have actually discussed the selection of LLMs and the impact of different relation extraction methods in the article. Specifically, in Section 7.1, our experiments focused on the influence of LLMs of different sizes on the performance of idea evaluation. We found that in many cases, using smaller LLMs not only reduces costs but also performs comparably to larger models. Based on these discussions, we also adopted the setting of smaller LLMs in the viewpoint extraction part of GraphEval. Additionally, we discussed the impact and selection of relation extraction methods from the perspective of edge density in Section 3. Indeed, because our method performs well and is robust, we did not extensively discuss specific parameter tuning and model design during the experiments.
> > >
> > > On the other hand, following the advice of other reviewers, we have added some discussions on ablation study in the ***[Appendix F]*** of our revised version. Specifically, the impact of varying edge densities is detailed in ***[Appendix F.1 and Table 22]***. Additionally, the effects of various lightweight graph neural network architectures are explored in ***[Appendix F.2 and Table 23]***. Furthermore, we evaluate the comparative impact of alternative relation extraction methods in ***[Appendix F.3 and Table 24]***.
> > >
> > > **Q12. The construction of the graph relies solely on textual information. Evaluating a paper’s acceptance potential based exclusively on textual relevance is not entirely reasonable.**
> > >
> > > **Response:** Thanks for the reviewer’s feedback. Indeed, as discussed in ***[lines 36-45]*** of our paper, the core focus of our article is on LLM-based idea evaluation, a promising hot topic in academia. Thus, similar to many existing works (Lu et al., 2024; Baek et al., 2024; Ubonsiri, 2024; Du et al., 2024; Lin et al., 2023a), we concentrate on using textual information to build our model. Additionally, in Section 3 of our paper, we discuss how different methods of viewpoint extraction can impact model performance and costs. We found that the method of textual relevance for viewpoint extraction not only significantly reduces the cost of extracting viewpoints using LLMs but also achieves effective results, which has been repeatedly verified in our experiments.

---

> > > > ### Author Response · Authors · 2024-11-24
> > > > **Could you let us know if our rebuttal has sufficiently addressed your concerns?**
> > > >
> > > > Dear Reviewer oRRW,
> > > >
> > > > We recognize that the timing of this discussion period may not align perfectly with your schedule, yet we would greatly value the opportunity to continue our dialogue before the deadline approaches.
> > > >
> > > > We hope that our responses and additional experiments have effectively addressed your concerns. We truly appreciate all the valuable advice we have received, and we are pleased to share that one of the reviewers has kindly recognized our improvements by raising their score. This acknowledgment reflects the positive impact of our collaborative efforts in enhancing the quality of the paper.
> > > >
> > > > Could you let us know if your concerns have been adequately addressed? If you find that your concerns have been resolved, we would appreciate it if you could reconsider the review score.
> > > >
> > > > Thanks!

---

> > > > ### Comment · Reviewer_oRRW · 2024-11-25
> > > > **Response to rebuttal**
> > > >
> > > > Thank you for addressing my comments.
> > > >
> > > > The authors have made efforts to respond to each of my concerns, but I believe there is still room for improvement to substantiate the effectiveness of the proposed approach. My primary concern remains the small size of the dataset used in the evaluation, which limits the generalizability of the findings.
> > > >
> > > > I have updated my evaluation accordingly but will maintain my recommendation for this paper. I will participate in the discussion with other reviewers and the rebuttal will be considered during the discussion.

---

> > > > > ### Author Response · Authors · 2024-11-27
> > > > > **Further response to Reviewer oRRW’s concern on scalability generalization**
> > > > >
> > > > > Thanks for the reviewer’s feedback. We want to reclaim the rationality of the settings of using our current dataset. As mentioned in our response to Q6, our paper primarily aims to demonstrate the technical contributions of GraphEval, and to ensure a fair comparison with LLM baselines, we have chosen the current dataset for our experiments. However, we follow the reviewer's suggestion and have conducted experiments on a large-scale dataset. Specifically, we conducted experiments on the ASAP-Review dataset [1]. The ASAP-Review dataset is an open peer review dataset that includes 5,192 ICLR papers from 2017-2020 obtained through OpenReview and 3,685 NeurIPS papers from 2016-2019 accessed through NeurIPS Proceedings. A detailed introduction to this dataset, along with its composition, can be found in Section 3.1 and Table 2 of [1]. Similar to the settings described in Section 6 of our paper, we used the abstracts of all papers in the dataset as inputs and the review decisions of the papers as the predicted labels, which included Accept (Oral), Accept (Spotlight), Accept (Poster), and Reject. We divided the dataset into training, validation, and test sets in the proportions of 70%, 10%, and 20%, respectively. It is important to note that for NeurIPS papers, since only accepted papers are included and no specific labels such as Oral, Spotlight, or Poster and ratings are provided, we have to assign all paper labels as Accept (Poster). This approach ensures the accuracy of the dataset because over 85% of the papers accepted at the NeurIPS conference are designated as posters. As shown in the table below, we compared the performance of GraphEval-GNN with that of Fine-tuned BERT and Prompted LLM on this dataset. We observed that GraphEval-GNN still maintains the best performance on this large-scale dataset, with an accuracy 9.8% better than the strongest baseline, Fine-tuned BERT. Furthermore, although the rare labels of Accept (Oral) and Accept (Spotlight) (less than 4%) make it difficult for all methods to perform well in terms of macro F1 score, GraphEval-GNN still achieved an 8% improvement in macro F1 score compared to Fine-tuned BERT. These observations demonstrate the robust generalization capability of GraphEval-GNN on large-scale datasets. All the discussions and experimental results mentioned above have been updated in Section G of the Appendix and Table 25 of the revised PDF version.
> > > > >
> > > > > Could you let us know if your concerns have been adequately addressed? If you find that your concerns have been resolved, we would appreciate it if you could reconsider the review score.
> > > > >
> > > > > Thanks!
> > > > >
> > > > > **Comparative Performance Results for Different Models on the ASAP-Review Dataset.** Bold text denotes the best results. For all metrics—Accuracy, Macro Precision, Macro Recall, and Macro F1 Score—higher values indicate more precise predictions.
> > > > >
> > > > > | Model               | Accuracy | Precision | Recall  | F1 Score |
> > > > > |---------------------|----------|-----------|---------|----------|
> > > > > | Prompted LLM (7B)   | 22.00%   | 11.04%    | 28.57%  | 12.83%   |
> > > > > | Prompted LLM (72B)  | 4.00%    | 4.00%     | 17.86%  | 3.04%    |
> > > > > | Finetuned-Bert      | 61.17%   | 29.81%    | 30.37%  | 29.86%   |
> > > > > | **GraphEval-GNN**   | **67.02%** | **33.11%** | **32.86%** | **32.20%** |
> > > > >
> > > > > **[1]** Yuan, Weizhe, Pengfei Liu, and Graham Neubig. "Can we automate scientific reviewing?." Journal of Artificial Intelligence Research 75 (2022): 171-212.

---

> > > > > > ### Author Response · Authors · 2024-11-28
> > > > > > **Could you let us know if our further response has sufficiently addressed your concerns?**
> > > > > >
> > > > > > Dear Reviewer oRRW,
> > > > > >
> > > > > > We hope that our further responses and additional experiments have effectively addressed your concerns. We truly appreciate all the valuable advice we have received, and we are pleased to share that other reviewers have kindly recognized our improvements by significantly raising their scores. This acknowledgment reflects the positive impact of our collaborative efforts in enhancing the quality of the paper.
> > > > > >
> > > > > > Could you let us know if your concerns have been adequately addressed? If you find that your concerns have been resolved, we would appreciate it if you could reconsider the review score.
> > > > > >
> > > > > > Thanks!

---

> > > > > > > ### Author Response · Authors · 2024-11-30
> > > > > > > **Looking forward to your reply**
> > > > > > >
> > > > > > > Dear Reviewer oRRW,
> > > > > > >
> > > > > > > We would like to express our sincere appreciation for your positive opinions and constructive review of our paper on the occasion of Thanksgiving. We apologize for intruding during your busy schedule, but as the discussion period is near its end, we would like to ensure our response aligns with your expectations and addresses your concerns. In our further response, we have **validated the good scalability and generalization capabilities of GraphEval-GNN on the large-scale ASAP-Review dataset**. We would like to know if your concerns have been adequately addressed. If you find that your concerns have been resolved, we would appreciate it if you could reconsider the review score.
> > > > > > >
> > > > > > > Wishing you a joyful Thanksgiving,
> > > > > > >
> > > > > > > Best regards,
> > > > > > >
> > > > > > > Authors

---

> > > > > > > > ### Author Response · Authors · 2024-12-02
> > > > > > > > **A friendly reminder for further discussion**
> > > > > > > >
> > > > > > > > Dear Reviewer oRRW,
> > > > > > > >
> > > > > > > > We hope this message finds you well. The rebuttal phase ends today and we would like to know if our further response has completely addressed your concerns. **In our further response, we have validated the good scalability and generalization capabilities of GraphEval-GNN on the large-scale ASAP-Review dataset.** We believe that we have addressed all of your previous concerns. We would really appreciate that if you could check our response and revised manuscript. If you find that your concerns have been resolved, we would appreciate it if you could reconsider the review score. Looking forward to hearing back from you.
> > > > > > > >
> > > > > > > > Best Regards,
> > > > > > > >
> > > > > > > > Authors

---

> > > > > > > > > ### Author Response · Authors · 2024-12-03
> > > > > > > > > **Kindly inquiry to revisit our responses and reconsider the score in light of the detailed clarifications and experiments provided**
> > > > > > > > >
> > > > > > > > > Dear Reviewer oRRW,
> > > > > > > > >
> > > > > > > > > We recognize that the timing of this discussion period may not align perfectly with your schedule, but we greatly value the opportunity to discuss with you, and we sincerely thank you for your valuable feedback and suggestions. **In our further response, we have validated the strong scalability and generalization capabilities of GraphEval-GNN on the large-scale ASAP-Review dataset.** We have dedicated significant time and effort to completing this experiment and hope to earn your recognition and further evaluation to address the concerns and improve the quality of our paper. We believe that we have addressed all of your previous concerns. We would greatly appreciate it if you could review our response and the revised manuscript. If you find that your concerns have been resolved, we kindly request that you reconsider the review score.
> > > > > > > > >
> > > > > > > > > Best Regards,
> > > > > > > > >
> > > > > > > > > Authors

---

### Official Review · Reviewer_zpR7 · 2024-11-02

**Soundness:** 4
**Presentation:** 4
**Contribution:** 3
**Rating:** 8
**Confidence:** 5

**Summary:**

This paper aims to develop a graph-based LLM framework for evaluating research ideas. Specifically, the authors first break down a whole idea into multiple viewpoints (nodes) and then connect them (making edges) based on their embedding-level similarity as well as the relation extraction approach, from which the authors construct the viewpoint graph. Then, by utilizing this viewpoint graph through simple label propagation or GNNs, the proposed approach predicts the quality of the research ideas. In addition to this, the authors propose the simple trick to evaluate the novelty of the ideas, by generating the labels for them and then training the GNN with them. The authors show that the proposed approach can not only predict the quality of the research ideas better than existing methods but also it can predict the novelty of the ideas.

**Strengths:**

* The proposed approach of evaluating the research ideas over the graph-structure (by breaking down them into multiple sub-ideas connected with other sub-ideas) is novel and sound.
* The proposed approach is superior to other research idea evaluation methods.
* This paper is very well-written and easy to follow.

**Weaknesses:**

I don't see any major weaknesses. But, there are some points that can make this paper more solid:
* The authors mainly evaluate the proposed method on only the research idea evaluation task, and, while this task is very important and less explored, this point may limit its generalizability to other tasks and domains (i.e., I believe it can be well applicable to other tasks related to evaluating long text and it is worth trying it).
* The authors can discuss some works on evaluating the long-form text [1, 2], as some of them tend to divide the long text into multiple subsets (similar to the proposed approach) and evaluate each of them.
* The authors can incorporate more analyses, to showcase the efficacy of the proposed approach more. One experiment that is worthwhile to try is, to showcase the generalizability of the proposed approach (or the constructed viewpoint graph) to new papers in 2023 by training it with papers before 2022.

---

[1] FActScore: Fine-grained Atomic Evaluation of Factual Precision in Long Form Text Generation, EMNLP 2023.

[2] Let's Verify Step by Step, arXiv 2023.

**Questions:**

Please see the weaknesses above.

---

> ### Author Response · Authors · 2024-11-21
> **Response to Reviewer zpR7**
>
> **Q1. The authors mainly evaluate the proposed method on only the research idea evaluation task, and, while this task is very important and less explored, this point may limit its generalizability to other tasks and domains (i.e., I believe it can be well applicable to other tasks related to evaluating long text and it is worth trying it).**
>
> **Response:**  Thanks for the reviewer’s constructive feedback. We appreciate the reviewer's suggestions, which are excellent for exploring the generalization capability of our method across various tasks and domains.
>
> Therefore, following the reviewer's advice, **we have chosen to conduct experiments on the dataset described in [1]**. Specifically, we used human-annotated data from the FActScore dataset, where each entry contains "atomic facts" about celebrities generated by LLMs, along with assessments from human annotators on whether these "atomic facts" were supported by the materials provided to the annotators. Based on the "atomic facts" and human annotations from the training set, our method needed to predict the labels of "atomic facts" in the test set. We selected topics such as Ramesses IV, Lanny Flaherty, and Florencia Bertotti, and divided the training, validation, and test sets in a 7:1:2 ratio. We compared GraphEval and some applicable baselines on this dataset in the following table. The experimental results in the table verify that **our approach performs well on the long-form text evaluation task, demonstrating good adaptability to various tasks.** All the discussions and experimental results mentioned above have been updated in ***[Section E.1 of the Appendix]*** and ***[Table 20]*** of the revised PDF version.
>
> **Comparative performance results on the Fact Verification dataset.** Bold text denotes the best results. For all metrics—Accuracy, Macro Precision, Macro Recall, and Macro F1 Score—higher values indicate more precise predictions.
>
> | Model              | Accuracy | Precision | Recall | F1 Score |
> |--------------------|----------|-----------|--------|----------|
> | Prompted LLM (7B)  | 49.79%   | 57.19%    | 52.27% | 47.59%   |
> | Prompted LLM (72B) | 59.52%   | 63.13%    | 60.35% | 56.33%   |
> | Finetuned-Bert     | 70.27%   | 69.74%    | 68.54% | 68.64%   |
> | GraphEval-LP       | 82.83%   | 83.41%    | 83.04% | 82.40%   |
> | **GraphEval-GNN**  | **85.00%** | **90.00%** | **83.00%** | **84.00%** |
>
> **Q2. The authors can discuss some works on evaluating the long-form text [1, 2], as some of them tend to divide the long text into multiple subsets (similar to the proposed approach) and evaluate each of them.**
>
> **Response:** Thanks for the reviewer’s insightful feedback. Following the reviewer’s suggestion, we add the corresponding citations and discussion in ***[Section 2]***:“Recently, some research works have been evaluating long-form texts, such as biographies of people (Min et al., 2023) and complex mathematical reasoning texts (Lightman et al., 2023). These studies divide the long text into multiple subsets and evaluate each of them. Inspired by these works, we decompose the obscure ideas into simple, understandable viewpoint nodes using LLMs, and further evaluate the idea based on graph algorithms.”.
>
> **Q3. The authors can incorporate more analyses, to showcase the efficacy of the proposed approach more. One experiment that is worthwhile to try is, to showcase the generalizability of the proposed approach (or the constructed viewpoint graph) to new papers in 2023 by training it with papers before 2022.**
>
> **Response:** Thank you for your insightful suggestions. We believe that this suggestion, like Q1, can effectively help validate the generalization ability of our method across different dimensions.
> Following the reviewer's advice, **we selected papers from before 2022 in the ICLR Papers dataset as the training and validation sets, and papers from 2023 as the test set.** We compared the performance of GraphEval with other classic baselines in the following table. **The results in the table validate GraphEval's temporal generalization ability in the task of idea evaluation.** All the discussions and experimental results  have been updated in ***[Section E.2]*** of the Appendix and ***[Table 21]*** of the revised PDF version.
>
> **Comparative performance results under the setting of idea evaluation of different years.** Bold text denotes the best results. For all metrics—Accuracy, Macro Precision, Macro Recall, and Macro F1 Score—higher values indicate more precise predictions.
>
> | Model             | Accuracy | Precision | Recall | F1 Score |
> |-------------------|----------|-----------|--------|----------|
> | Prompted LLM (7B) | 16.67%   | 20.63%    | 26.12% | 18.25%   |
> | Prompted LLM (72B)| 14.29%   | 11.25%    | 32.47% | 11.76%   |
> | Finetuned-Bert    | 48.41%   | 42.46%    | 36.14% | 31.57%   |
> | GraphEval-LP      | 63.20%   | 52.38%    | 48.60% | 44.72%   |
> | **GraphEval-GNN** | **76.19%** | **48.25%** | **57.38%** | **51.32%** |

---

> > ### Author Response · Authors · 2024-11-24
> > **Could you let us know if our rebuttal has sufficiently addressed your concerns?**
> >
> > Dear Reviewer zpR7,
> >
> > We recognize that the timing of this discussion period may not align perfectly with your schedule, yet we would greatly value the opportunity to continue our dialogue before the deadline approaches.
> >
> > We hope that our responses and additional experiments have effectively addressed your concerns. We truly appreciate all the valuable advice we have received, and we are pleased to share that one of the reviewers has kindly recognized our improvements by raising their score. This acknowledgment reflects the positive impact of our collaborative efforts in enhancing the quality of the paper.
> >
> > Could you let us know if your concerns have been adequately addressed? If you find that your concerns have been resolved, we would appreciate it if you could reconsider the review score.
> >
> > Thanks!

---

> > > ### Comment · Reviewer_zpR7 · 2024-11-30
> > >
> > > Thank you for your response. The authors clearly address my concerns and (after reading other reviews and responses as well) I will keep my score of 8: accept, good paper.

---

> > > > ### Author Response · Authors · 2024-11-30
> > > > **Thanks for the Reviewer’s Constructive Feedback**
> > > >
> > > > Thank you for your thoughtful and constructive feedback. We are pleased to hear that our responses have addressed your concerns. We are committed to incorporating the suggested changes in our revisions to further enhance the manuscript. By the way, we also sincerely wish you a Happy Thanksgiving. Thank you for the valuable feedback and hard work you provided during the rebuttal phase.

---

### Comment · Area_Chair_Sya7 · 2024-11-25
**Please reply to the authors' response.**

Dear reviewers,

The ICLR author discussion phase is ending soon. Could you please review the authors' responses and take the necessary actions? Feel free to ask additional questions during the discussion. If the authors address your concerns, kindly acknowledge their response and update your assessment as appropriate.


Best,
AC

---

### Meta-Review · Area_Chair_Sya7 · 2024-12-17

**Metareview:**

The paper introduces a novel framework that leverages graph structures to enhance the evaluation of research ideas using large language models (LLMs). The framework, which includes GraphEval-LP and GraphEval-GNN, demonstrates improvements in robustness and accuracy, outperforming baseline methods by a large margin.

Reviewers are very positive about the paper in general. However, they have also pointed out some limitations, such as the small dataset size used for evaluation, which may affect the generalizability of the results, and concerns about the scalability of GraphEval-GNN for larger datasets. The paper also lacks a thorough theoretical analysis of the proposed mechanisms. Despite these issues, the innovative approach and significant performance improvements make this paper a valuable contribution to the field. The rebuttal includes some new results which are suggested to be included to the paper.

Therefore, I recommend accepting.

**Additional Comments On Reviewer Discussion:**

The authors addressed most concerns of the reviewers. One reviewer was not actively involved in the rebuttal process. The reviewer's only concern remains the small size of the dataset used in the evaluation. The authors have provided reasonable response to the concerns.

---

### Decision · Program_Chairs · 2025-01-22

Accept (Poster)